

# Modification of the RothC model to simulate soil C mineralization of exogenous organic matter

Claudio Mondini[1], Maria Luz Cayuela[2], Tania Sinicco[1], Flavio Fornasier[1], Antonia Galvez[3], Miguel Angel Sánchez-Monedero[2]

[1]Consiglio per la Ricerca in Agricoltura e l'Analisi dell'Economia Agraria (CREA), Research Group of Gorizia, Via Trieste 23, I-34170 Gorizia, Italy.
[2]Centro de Edafología y Biología Aplicada del Segura (CEBAS-CSIC), Department of Soil and Water Conservation and Waste Management, Campus Universitario de Espinardo, Apartado de Correos 164, E-30100 Espinardo, Murcia, Spain.
[3]Instituto Andaluz de Ciencias de la Tierra (CSIC-UGR), Avenida de las Palmeras 4, E-18100 Armilla, Granada, Spain.

*Correspondence to: Claudio Mondini (claudio.mondini@crea.gov.it)*

**Abstract.** The development of soil organic C (SOC) models capable to produce accurate predictions of the long term decomposition of exogenous organic matter (EOM) in soils is important for an effective management of organic amendments. However, reliable C modelling in amended soils requires specific optimization of current C models to take into account the high variability of EOM origin and properties. The aim of this work was to improve the prediction of C mineralization rates in amended soils by modifying the RothC model to encompass a better description of EOM quality.

The standard RothC model, involving C input to the soil only as decomposable (DPM) or resistant (RPM) organic material, was modified by introducing additional pools of decomposable (DEOM), resistant (REOM) and humified (HEOM) EOM. The partitioning factors and decomposition rates of the additional EOM pools were estimated by model fitting to respiratory curves of amended soils. For this task, 30 EOMs from 8 contrasting groups (compost, anaerobic digestates, sewage sludges, agro-industrial wastes, crop residues, bioenergy by-products, animal residues, meat and bone meals), were added to 10 soils and incubated under different conditions.

The modified Roth C model was fitted to C mineralization curves in amended soils with great accuracy (mean correlation coefficient: 0.995). Differently to the standard model, the EOM-optimized RothC was able to better accommodate the large variability in EOM source and composition, as indicated by the decrease in the root mean squared error of the simulations for different EOMs (from 29.9% to 3.7% and from 20.0% to 2.5% for bioethanol residue and household waste compost amended soils, respectively). Average decomposition rates for DEOM and REOM pools were 89 $y^{-1}$ and 0.4 $y^{-1}$, higher than the standard model coefficients for DPM (10 $y^{-1}$) and RPM (0.3 $y^{-1}$).

Results indicate that explicit treatment of EOM heterogeneity enhances the model ability to describe amendment decomposition under laboratory conditions and provides useful information to improve C modelling on the effects of different EOM on C dynamics in agricultural soils.

Future researches involve the validation of the modified model with field data and its application to long term simulation of SOC patterns in amended soil at regional scale under climate change.



# 1 Introduction

Exogenous organic matter (EOM) is all organic material of biological origin that is applied to cultivated fields for the
purpose of growing crops, improving soil quality and restoring or reclaiming land for future use (Marmo et al., 2004).
Agricultural utilization of EOM is considered to be an effective way of restoring losses of soil organic matter (SOM) and
offsetting soil degradation and climate change (Lal, 2004; Smith, 2004a; Smith, 2004b). A reliable management of
amendment requires a thorough knowledge of EOM mineralization patterns, as the rate of EOM decomposition is critical to
determine its effects on soil properties, nutrient cycling and C accumulation.

However, prediction of EOM transformation in soil is a very difficult task as EOM mineralization is an extremely complex
process depending on several factors such as EOM biochemical composition, EOM stabilization treatment, size and activity
of soil microorganisms and pedoclimatic conditions (Franzluebbers, 2004). In particular, EOM composition is extremely
variable, since organic residues may have plant or animal origin and may have undergone different stabilization treatments.

Process-oriented soil organic C (SOC) modelling represents a reliable solution for an efficient management of EOM
amendment as it offers in principle a unique mean of addressing the high variability in the properties of EOM and
pedoclimatic conditions and the complexity of mechanisms and factors affecting C mineralization in the field. The
effectiveness of models in predicting long term C changes in amended soils has been recently supported by findings of
Karhu et al. (2012), Noirot-Cosson et al. (2016), Peltre et al. (2012), and Plaza et al. (2012), who found good correlations
between modelled and measured C stocks for different types and amounts of EOM. Some examples of C models that have
been utilized to simulate SOC trends in amended soils at field scale are reported in Table 1.

As the composition and properties of amendments is the most important factor controlling their decomposition (Cavalli et
al., 2014; Do Nascimento et al., 2012; Karhu et al., 2012), several authors have highlighted the importance of a proper
characterization of EOM to decrease the uncertainty in model predictions of SOC trends in amended soils. Regarding the
relevance of organic matter (OM) quality in SOC modelling, most models are based on the concept that decomposition can
be adequately simulated by assuming different conceptual or functional pools of OM that decay according to first order
kinetics with specific decomposition rate constants (Borgen et al., 2011). Exogenous organic matter is composed by
substances with different properties and distinct levels of accessibility to microorganisms. The rate of EOM mineralization is
mainly determined by the combination between quality and accessibility and the intensity of response to environmental
factors of substrates with diverse characteristics. Therefore, an accurate partitioning of EOM into a number of discrete pools
and estimation of their functional characteristics (i.e. initial C and N contents, decomposition rate) is of great importance to
improve model predictions (Sierra et al., 2011; Thuries et al., 2001). Generally C models identify two or three pools of
EOM, while their decomposition rates can be fixed or variable according to the specific EOM. However, rigorous methods
for establishing entry pools that account for the diversity of EOM have not been developed to date (Peltre et al., 2012). This
represents one of the major problems for a reliable C modelling of amended soil, as this separation is challenging and no



universally recognized methodology exists to perform this task. According to Petersen et al. (2005b), the uncertainty related to the fractionation of EOM into pools is one of the major weaknesses associated to the C modelling of amended soils.

Several approaches have been proposed in order to determine EOM pools partitioning factors and decomposition rates, but, to date, no satisfactory method for such characterization have been found. The main approaches that have been devised so far are based on chemical or kinetic subdivision of EOM. Partitioning based on the chemical properties of EOM is generally

performed by stepwise chemical digestion (SCD) and near infrared reflectance spectroscopy (NIRS)(Borgen et al., 2011; Peltre et al., 2011). Such methods are relatively rapid and simple, but presents the main disadvantage that these operationally defined fractions do not precisely correspond to the model pools. An alternative to chemical analysis is to characterize EOM pools by direct fitting of simulated $CO_2$ emissions to measured respiration curves from incubation experiments (Barak et al., 1990). Fitting pool parameters in this way provides kinetically defined parameters that reflect the rate of C mineralization

observed for each residue (Borgen et al., 2011; Trinoustrot et al., 2000) and is appealing because it allows for simultaneous estimation of both pool size and decomposition rate (Scharnagl et al., 2008) that can be directly used in process-oriented models (Batlle-Aguilar et al., 2011). EOM pools characterization by fitting $CO_2$ respiration from incubations was successfully achieved for NCSOIL (Corbeels et al., 1999; Gabrielle et al., 2004; Noirot-Cosson et al., 2016), CANTIS (Garnier et al., 2003; Parnaudeau, 2005) and TAO (Pansu and Thuries, 2003) models and was also performed by several

other researchers (Antil et al., 2011; Borgen et al., 2011; Cavalli and Bechini, 2011). In general, results of previous works on EOM characterization for soil C model calibration showed that kinetically defined partitioning enhances the predictions of mechanistic models compared to operationally defined fractions (Borgen et al., 2011; Gabrielle et al., 2005) and that the wider applicability of EOM characterization by SCD and NIRS is obtained at the expense of a lower accuracy.

To date, there are no soil C models specifically developed to evaluate the C accumulation potential of amended soils, with

the only exception of the TAO (transformation of added organic matter) model (Pansu and Thuries, 2003). Furthermore, C models have not been extensively calibrated in amended soils and the quality of organic inputs is an aspect that has not been adequately considered and needs further investigation (Parshotam et al., 2001). An example of this inadequacy is represented by the Rothamsted Carbon model (RothC), one of the most well-known and widely used models simulating SOC trends (Jenkinson et al., 1991; McGill, 1996), because it requires relatively few and easily available parameters and input data.

Although it has also been used in a few occasions to make predictions following application of EOM (Yokozawa et al., 2010), its actual structure suggests that the model is not particularly suited for C simulation in amended soils. Carbon inputs to the model are divided in the decomposable plant material (DPM) and resistant plant material (RPM) pools, each one characterized by a specific decay rate. This implies that the model does not allow C inputs deriving from crop residues to be differentiated from EOM. Secondly, the quality of the OM entering in the soil is only defined by the partitioning between

decomposable and resistant organic materials, as the decomposition rate are fixed and constant for each pool. The insensitivity of the actual model to the variation in the quality of inputs was showed by Falloon (2001). In fact, RothC allows only a specific EOM, namely farmyard manure, to be treated separately from crop residues, but its partition coefficients and



decomposition rates are fixed. This model behaviour contrasts with the large variability in the decomposition rate of different EOMs.

The aim of this study was to devise an easy and effective procedure for the optimization of the RothC model to improve the prediction of EOM-C mineralization, as a first step of model development for reliable SOC simulation in amended soils. Such procedure is based on two steps:

- modification of RothC involving the introduction of additional entry pools of EOM
- utilization of information derived from laboratory incubation experiments to define the size and decomposition rate of the additional EOM pools.

## 2 Materials and Methods

### 2.1 Incubation experiments

#### 2.1.1 Soils used for incubations

The soils used for the incubation experiments were sampled from agricultural areas in the Mediterranean area and specifically in Northern Italy and Southern Spain. The soils were sampled at 5-20 cm depth with an auger and several subsamples were pooled together to obtain a representative sample. Location and main physico-chemical characteristics of the soils are reported in Table 2.

The soil were sieved moist through a 2 mm aperture grid and stored (5 °C) until the beginning of the experiments. Before the starting of the trials, the soils were pre-conditioned by incubation under aerobic conditions for 7 days at the same temperature and water content adopted for the experiments.

The range of soils showed a widely different texture and pH. Apart from Gorizia, Bueriis and Lodi, the samples were characterized by low contents of organic C and N and a small pool of soil microbial biomass.

#### 2.1.2 EOMs used for incubations

As a whole 30 different EOMs were utilized for the incubation experiments. They were considerably distinct in terms of origin, chemical composition, and stabilization/transformation processes to which they were subjected. According to the above properties they were classified in 9 different EOM groups (Table3) and their main features and properties are reported on Table 4. Most of them presented an alkaline pH, while the organic wastes with a pH < 5.2 were bioethanol residue, hydrolyzed leather and two-phase olive mill waste. The total organic C (TOC) concentration ranged between 28.2% and 53.0%, except for green waste biochar, which had a TOC content of 86.0%. Total N varied between 0.3% and 17%, mainly depending to the EOM origin. Generally, vegetal derived EOM as vine shoots compost, household waste compost, green waste compost, crop residues, two-phase olive mill waste and green waste biochar showed low levels of total N (0.3-2.3%). On the other side, EOM of animal origin (meat and bone meals, blood meal, horn and hoof meal) showed high values of N



(8.2-17.0%). As a consequence of the variability in C and N content, the C/N ratio ranged between 3 (horn and hoof meal) to almost 200 (wheat straw) and 345 (green waste biochar). The differences among EOMs were also highlighted by the content

of easily available C (WSC) and N (WSN), varying from 0.1 to 203 g kg$^{-1}$ and from 0 to 37.9 g kg$^{-1}$ for WSC and WSN, respectively. The EOMs showing the highest contents of easily degradable C and N were bioethanol residue and blood meal. In general high concentrations of mineral N ($NO_3^-$ and $NH_4^+$) were found for liquid digestates. Conversely, bioenergy by-products two-phase olive mill waste, and biochar were characterized by very low amounts of $NO_3^-$.

### 2.1.3 Amended soils incubation experiments

The solid residues were ground and sieved (< 0.5 mm) to homogenize their particle size before application. The residues were thoroughly mixed with pre-conditioned moist soil samples (50 g dry weight basis) at the beginning of the incubation and kept under aerobic conditions in the dark in 130 ml plastic jars in a thermostatic chamber. In the case of liquid residues soil were pre-incubated at such humidity that after EOM addition they were brought to the moisture content required for incubation. Unamended soils were also included as a control. Each treatment was replicated at least twice. The moisture

levels in the jars were checked weekly by measuring weight loss, and deionised water was added when necessary to maintain constant moisture. Incubations were performed for a range of temperature (10-30 °C), soil moisture (20-40% water holding capacity (WHC)) EOM rate (0.1-0.75 %) and time (7-37 d) conditions. Details on incubation conditions (soil type, rate of residue, soil water content, temperature and incubation time) are reported on Tables S1-S6 of Supplement. As a whole, more than 30 incubations (each involving 7 treatments) were performed, utilizing 30 different residues and 10 soils with

contrasting properties for a total of 224 treatments.

### 2.1.4 Soil $CO_2$ measurement

$CO_2$ evolution was measured every 6 h on aliquots of moist soils by means of an automated system for gas sampling and measurement (Mondini et al., 2010). (Fig. 1). The 'apparent' net C mineralisation (C derived from the residues) was calculated as the difference between the $CO_2$-C emitted by the EOM amended soil and that produced over the same period

by the unamended control soil.

## 2.2 RothC model modification and optimization

### 2.2.1 Description of the RothC model

The Rothamsted Carbon model (RothC) was one of the first multi-compartmental models to be developed (Coleman and Jenkinson, 1996; Jenkinson and Rayner, 1977) and has been evaluated and optimized for a variety of ecosystems including

155 croplands, grasslands and forests (Coleman et al., 1997; Falloon and Smith, 2002; Smith et al., 1997) and in various climate regions, including Mediterranean and semi-arid environments (Farina et al., 2013; Francaviglia et al., 2012; Skjemstad et al., 2004).



RothC describes the dynamics of SOM by splitting it into five compartments with different decomposition (or kinetic) rate constant (K), namely decomposable plant material (DPM, $K = 10$ $y^{-1}$), resistant plant material (RPM, $K = 0.30$ $y^{-1}$), soil microbial biomass (BIO, $K = 0.66$ $y^{-1}$), humified organic matter (HUM, $K = 0.02$ $y^{-1}$) and inert organic matter (IOM). Each compartment, except IOM, follows first-order decay kinetics, i.e. each pool is considered well-mixed and chemically homogeneous and the decomposition rate is assumed to be controlled by the available substrate. The proportion of organic matter decomposed per unit time is therefore constant and equal to K.

Carbon inputs to the soil are divided into compartments of DPM and RPM with partitioning factors (f) depending on the nature of the inputs. At each monthly time step, part of each C input pool is decomposed according to its specific decomposition rate. Part is mineralized as $CO_2$ and the rest is transferred to the compartments BIO and HUM. The proportion of the decomposed pool converted to $CO_2$ and (BIO + HUM) is determined by the clay content of the soil. The rate constants are modified at each period by three multipliers, depending on the temperature, the moisture deficit of soil and the presence/absence of vegetation. Due to extensive previous evaluations of model performance (e.g. Smith et al., 1997), no further validation of the current model is presented here.

### 2.2.2 Modification of the RothC model

The standard model structure considers the C input to the system only in terms of decomposable (DPM) and resistant (RPM) organic material, i.e. each pool comprises both crop residues and the possibly added EOM. Furthermore the quality of the incoming materials to the soil can only be described by varying the partitioning factor between DPM and RPM (i.e. the DPM/RPM ratio), as the decomposition rate of each pools is constant. The only EOM that the model can treat separately from crop residues is farmyard manure, but the partition coefficients and decomposition rates of farmyard manure pools are fixed and cannot be altered by the user. In order to allow the model to separately consider crop residues and EOM and for a better description of amendment quality a modification to the model source code was performed involving the inclusion of two additional pools of EOM (decomposable EOM (DEOM) and resistant EOM (REOM)), each one characterized by specific and variable rate of decomposability. Furthermore, for some materials characterized by the presence of stable OM, a third EOM pool is introduced (humified EOM (HEOM)) which is directly incorporated into the soil HUM pool. EOM added to the soil is split into the DEOM, REOM and HEOM pools according to the partitioning factors $f_{DEOM}$, $f_{REOM}$ and $f_{HEOM} = 1 - f_{DEOM} - f_{REOM}$. DEOM and REOM pools decompose with specific decomposition rates ($K_{DEOM}$ and $K_{REOM}$), that may be different from those of plant residues, while HEOM, being directly incorporated into the HUM pool, decomposes with the same decomposition rate ($K = 0.02$ $y^{-1}$). Decomposed DEOM and REOM are split in $CO_2$, BIO and HUM. The proportion of decomposed DEOM and REOM that goes to $CO_2$, BIO and HUM is regulated in the same way as for the entry pools of plant residue. Compost, anaerobic digestates and agro-industrial wastes were the EOMs partitioned in 3 EOM pools in this study. Below is reported a mathematical representation of the modified model as a set of differential equations:

$$\text{dDPM/dt} \quad = \quad f_{DPM}P - K_{DPM}DPM \tag{1}$$



$$dRPM/dt \quad = \quad (1 - f_{DPM})P - K_{RPM}RPM \tag{2}$$

$$dDEOM/dt \quad = \quad f_{DEOM}E - K_{DEOM}DEOM \tag{3}$$

$$dREOM/dt \quad = \quad f_{REOM}E - K_{REOM}REOM \tag{4}$$

$$dBIO/dt \quad = \quad \alpha K_{DPM}DPM + K_{RPM}RPM + K_{DEOM}DEOM + K_{REOM}REOM + K_{HUM}(HUM +$$

$$(1 - f_{DEOM} - f_{REOM})E)) - (1 - \alpha)K_{BIO}BIO \tag{5}$$

$$dHUM/dt \quad = \quad \beta(K_{DPM}DPM + K_{RPM}RPM + K_{DEOM}DEOM + K_{REOM}REOM + K_{BIO}BIO)$$

$$- (1 - \beta)K_{HUM}(HUM + (1 - f_{DEOM} - f_{REOM})E) \tag{6}$$

where:

DPM = decomposable plant material; RPM = resistant plant material; HUM = humified organic matter; BIO = soil microbial biomass; DEOM = decomposable EOM; REOM = resistant EOM.

$f_{DPM}$ = partitioning factor for DPM; $f_{DEOM}$ = partitioning factor for DEOM; $f_{REOM}$ = partitioning factor for REOM.

$K_{DPM}$ = decomposition rate for DPM; $K_{RPM}$ = decomposition rate for RPM; $K_{BIO}$ = decomposition rate for BIO; $K_{HUM}$ = decomposition rate for HUM; $K_{DEOM}$ = decomposition rate for DEOM; $K_{REOM}$ = decomposition rate for REOM.

P = plant (crop residue) input; E = EOM input; $\alpha$ = transfer coefficient to BIO pool; $\beta$ = transfer coefficient to HUM pool.
The C flow of the standard and modified model is reported in Fig. 2.

### 2.2.3 Optimization of the modified RothC model

Cumulative respiration curves of 224 EOM amended soils incubated under different conditions were applied for parameter estimation. Biochar amended soils (n = 4) were excluded from the procedure of parameter estimation due to very low values of $CO_2$-C emissions resulting in not statistically robust respiration curves.

To enable model fitting of respiration data from the incubation trials, an Excel version of the RothC model (26.3 version) was utilized. The Excel version of the model was tested for correctness under several RothC standard scenarios.

All simulations were run as difference with the control treatment (i.e. only the $CO_2$ derived from EOM was simulated) utilizing a time step of 0.25 $d^{-1}$. Thus the initial size of the soil organic pools was virtually set to zero, including the size of the inert OM pool (IOM). This was possible because in the RothC model the C trend of each pool is described with first order kinetics. Hence, the fate of total soil C is the sum of the fate of the C of the different pools. Consequently, the difference in $CO_2$ evolution between soils with and without EOM application corresponds to the $CO_2$ derived from the additional input of OM in the soil. It was therefore assumed that the decomposition of humified SOM was unaffected by the decomposition of added residues (i.e. no priming effect was caused by EOM application to the soil).

Model fitting to measured values was conducted by changing individual partition coefficients and decomposition rate constants of the EOM pools by stepwise iteration using Excel-Solver with the Newton method until maximum agreement between measured and simulated amounts of $CO_2$ was achieved assuming as a criteria the smallest sum of squared residuals





(SSR). For each EOM and incubation conditions, an 'individual' fitting procedure was used to minimise the difference between observed and simulated values. The partitioning factors ($f_{DEOM}$, $f_{REOM}$, $f_{HEOM}$) and the decomposition rate constants ($K_{DEOM}$, $K_{REOM}$) of the different C pools of EOM were taken into account for inverse modelling. The 5 parameters were optimized simultaneously, considering the following constraints in order to avoid biologically unrealistic parameter estimates:

$f_{DEOM} + f_{REOM} + f_{HEOM} = 1$

$f_{HEOM} < 0.3$ for anaerobic digestates and agro-industrial wastes

$K_{REOM} > 0.15$ $y^{-1}$

$k_{DEOM} < 230$ $y^{-1}$

The partitioning factor for HEOM ($f_{HEOM}$) was set to a maximum of 0.3 in the case of digestate and agro-industrial wastes according to the values found by Cavalli and Bechini (2011; 2012) after model calibration for soil amended with pig slurries stored under anaerobic conditions before use. The minimum $K_{REOM}$ value was set at 0.15 $y^{-1}$ according to the RothC modification proposed by Skjemstad et al. (2004). $k_{DEOM}$ was set to a maximum of 230 $y^{-1}$ in agreement to maximum values found by Thuries et al. (2001) utilizing a 3 EOM pools model for 14 different plant residues, compost and manures. This constraint was not considered in the case of blood meal as the respiration curves presented a very steep initial phase, an indication of a decomposable pool characterized by high degree of decomposability. This is supported by results of Thuries et al. (2001) who found a decomposition rate constant of 243 $y^{-1}$ for the labile pool of animal residues.

The accuracy of the model to simulate C mineralization were assessed according to the criteria proposed by Smith et al. (1996), utilizing the worksheet MODEVAL 2.0 for Windows (Smith and Smith, 2007). The total difference between measured and simulated values, expressed as percentage of the mean observed values, was considered by calculating the root mean squared error (RMSE). The lower limit for RMSE is 0, which denotes no difference between measured and simulated values. The association between simulated and measured values (i.e. the percentage of total variance in the observed data that is explained by the predicted data) was evaluated by the sample correlation coefficient (R). The error in the simulation as a proportion of the measurement was evaluated by the relative error (E), expressed as the mean error percentage over all the measurements. The consistent errors or bias in the model was evaluated by the mean difference between measured and simulated data (M). Because M does not include a square term, simulated values above and below the measurements cancel out and so any inconsistent errors are ignored.

## 3 Results

### 3.1 EOM soil mineralization

As an example of the rate of $CO_2$ mineralization from soil amended with different EOMs, Fig. 3a shows the dynamics of $CO_2$ evolution from the Llano de la Perdiz soil. Range and mean values of net C mineralization for the different EOM groups, as defined in the materials and methods section, are reported in Table 5 which summarizes results from all the



incubations performed utilizing different conditions and incubations carried out under standard laboratory conditions (20 °C, 40% WHC, 0.5% application rate and 30 days incubation period).

Considering all the incubations performed, the extra $CO_2$-C varied in the range 0.01-38.6% of the added EOM-C (Table 5). According to mean values of net C mineralization obtained under standard laboratory conditions, the different EOM groups can be ranked as follows (values in parenthesis are the percentage of added C emitted as $CO_2$-C): biochars (0.02%) <

composts (3.0%) < anaerobic digestates (4.0%) < sewage sludges (4.8%) < agro-industrial wastes (6.3%) < crop residues (10.4%) < bioenergy by-products (12.8%) < animal residues (16.8%) < meat and bone meals (21.3%).

In the case of compost, EOM mineralization ranged from 0.9 to 11.1%, with a mean value of 3.7% (Table 5). The total extra $CO_2$-C evolved from the soils amended with meat and bone meals ranged between 7.8 and 38.6%, while in the case of bioenergy by-products net $CO_2$-C production was in the range 6.9-16.8%. Extremely low values of C mineralization (0.01-

0.04%) were recorded for biochar amended soil (Table 5). Significant relationship between cumulative net $CO_2$-C of different EOM group and chemical properties were found only for water soluble N ($r^2 = 0,70$; P<0.01).

### 3.2 Optimization of the modified model by incubation data

Mean, minimum and maximum values of statistical indicators utilized to evaluate the model goodness of fit between measured and simulated data are reported on Table 6. On Tables S1-S6 of Supplement, the pool parameters, the incubation

conditions and the statistical indicators of model goodness of fit are detailed. Biochar amended soils were omitted from the optimization procedure due to very low $CO_2$ emissions values resulting in not statistically robust respiratory curves.

As a whole, the modified model was able to fit very well the respiratory response of the amended soils, as indicated by the statistical indicators (Table 6; Tables S1-S6 of Supplement). The only exceptions were represented by some soil amended with a low dose of anaerobic digestate (100 kg N ha$^{-1}$). The mean correlation coefficient (R) for all incubations was 0.995

and was higher than 0.945 for all but one EOM. The root mean squared error (RMSE) for vine shoots compost, household waste compost and bioethanol residue was 4.3%, 2.5% and 3.7%, respectively, while considering all the cases it was 4.5%. The relative error (E) ranged between -16.4 and 3.5% (Tables S1-S6 of Supplement). The goodness of fit was also underlined by the very low values of M (on average -1.2 µg $CO_2$-C g$^{-1}$)(Table 6). As an example of curve fitting, Fig. 3b depicts measured and simulated net cumulative $CO_2$-C evolution for EOMs reported in Fig. 3a.

Average decomposition rate for EOM and REOM pools were 89 y$^{-1}$ and 0.4 y$^{-1}$. Evaluation of pool parameters showed a large variability in the composition and decomposition rates of the studied EOMs. Range of different parameters were: 0-0.63; 0.21-0.98; 0.06-0.78 for $f_{DEOM}$, $f_{REOM}$, $f_{HEOM}$ and 11-330; 0.15-2.51 for $K_{DEOM}$, $K_{REOM}$, respectively (Tables S1-S6 of Supplement). Coefficients of variation of the parameters considering all the treatments were 83, 24, 53, 69 and 95% for $f_{DEOM}$, $f_{REOM}$, $f_{HEOM}$, $K_{DEOM}$ and $K_{REOM}$, respectively. Pool sizes and decomposition rates were not significantly correlated.

No statistically significant relationships were found between pool parameters and chemical properties of different EOM groups. Partition coefficients for DEOM were significantly correlated with cumulative net $CO_2$-C ($r^2 = 0,92$; P < 0.01). Calculation of  mean pool parameters and associated percent of variation of standard error of all incubations performed with





the same EOM type (Table 7) or with the same EOM group (Table 8) showed always a relative standard error smaller than 50%, a threshold value proposed by Robinson (1985) for a statistically acceptable estimation of model parameter, with a single exception in the case of stable compost CMC VII (Table S1 of Supplement).

# 4 Discussion

## 4.1 EOM soil mineralization

The ranking of the different EOM groups according to mean values of net C mineralization was in agreement with results of similar studies on the decomposability of EOMs of different origin and nature (Lashermes et al., 2009; Thuries et al., 2001). Values of net C mineralization (expressed as percentage of added C) for compost amended soil were similar to the ones recorded by De Neve et al. (2003), who measured $CO_2$-C values in the range 1.8 - 8.8% of added C for different composts. Values of mean $CO_2$-C respiration of meat and bone meals (16.8%) are in agreement with other previous C mineralization studies of residues characterized by low values of C/N ratio, as, for example, 16% and 19% obtained from poultry manure and pig slurry after a 20-day incubation at 22 °C (Levi-Minzi et al., 1990). Regarding by-products from bioenergy production, values of C mineralization in the present study were significantly lower to the ones measured by Cayuela et al. (2010). This dissimilarity can be attributed to the different conditions utilized for the incubation. Nevertheless, the organic residues showed the same relative differences in $CO_2$ production. The significant correlation between EOM water soluble N and mineralized added C is in agreement with previous studies showing that N availability is an important factor regulating EOM decomposition (Trinsoutrout et al., 2000).

## 4.2 Model modification

The development and optimization of SOC models capable to produce accurate and reliable predictions of the long term decomposition of EOM in soils (Karhu et al., 2012) represents an essential prerequisite for their utilization as a tool for an effective management of EOM amendment. However, the standard RothC model is not properly suited for the simulation of soils amended with a range of contrasting EOMs, as it does not distinguish between crop residues and EOM, notwithstanding their widely different nature, and is not sensitive to the variation in the quality of inputs. This limitation of RothC is highlighted by the results of Tits et al. (2014), who simulated 30 years of compost addition. The quality of EOM in their work was addressed by calibrating the DPM/RPM ratio with the SOC content, however this ratio encompassed not only EOM quality, but also the quality of input materials (crop residues). Consequently, the calibrated DPM/RPM ratio was site specific, as this ratio depended not only from compost properties, but also by the crop type and management of the site utilized for calibration. The fact that the same pool structure is used to represent organic materials that widely differ in composition and decomposition pattern (e.g. crop residues vs. compost) simplifies model structure, but is likely to generate less accurate results (Cavalli and Bechini, 2011). The suggested model modification was proposed to overcome the limitation of the standard model in taking into account the variation in the quality of added organic C.





In the modified Roth C model, the introduction of 3 additional EOM pools, with the more stable one directly incorporated
into the humus pool of SOM (Fig. 2), is conceptually similar to that of the CNSIM model (Pedersen et al., 2007; Petersen et
al., 2005b). In this model, EOM is partitioned in 3 pools, namely metabolic (decomposable), structural (resistant) and
recalcitrant (humic-like EOM), with the latter directly incorporated into the soil humic pool. Petersen et al. (2002) indicated
that OM from slurry and farmyard manure is generally assumed to contribute relatively more to SOM build-up than plant
residues and this is modelled in the CNSIM model by transporting a fraction of added manure directly into the slow native
OM pool. This assumption is also in agreement with the results of Nicolardot et al. (2001), who found that NCSOIL model
tends to overestimate residue-derived $CO_2$ production and attributed this to the structure of the model that encompasses
assimilation of the whole residue by the soil microflora. The authors suggest that models allowing for direct incorporation
into stable SOM of a proportion of EOM-C (resistant compounds such as lignin), instead of microbial incorporation, are
more reliable for the simulation of long term SOM evolution. In the modified RothC model the characteristic of HEOM pool
in terms of degradability was considered to be similar to the soil HUM pool and, for this reason, it was attributed to HEOM
the same decomposition rate of HUM (K = 0.02 $y^{-1}$). This assumption was made on the basis that certain EOMs, such as
composts, anaerobic digestates and olive mill wastes, usually undergo a stabilization process that transforms part of the raw
material into humic-like substances resembling the properties of soil humic substances (Brunetti et al., 2012; Fakharedine et
al., 2006; Inbar et al., 1990).

The model modification proposed in the present study originated from preliminary tests showing the inaccuracy of the
standard model to simulate the respiration of amended soil incubated in laboratory. This is not surprising given the high
variability of chemical composition, origin and properties of EOM. The necessity to optimize RothC to adequately simulate
C dynamics in amended soil is supported by the results of Falloon (2001), who utilized either the standard and a modified
version of RothC to model C trends in a long term experiment (45 years) dealing with sewage sludge amendment. Using the
standard model version and treating the sewage sludge as an input of farmyard manure to soil resulted in a significant
underestimation of SOC increase. A modified version of the model was then used, assuming that sewage sludge was
composed of 10% DPM, 70% RPM and 20% HUM (rather than the standard values of 49% DPM, 49% RPM and 2% HUM
used for farmyard manure), and this resulted in a much closer agreement between modelled and measured SOC trends. The
specific partitioning factors for sewage sludge reflect the evidence that it is composed of a greater proportion of resistant and
humified organic material and less decomposable organic material compared to farmyard manure (McGrath and Brookes,
1986). Further support to the effectiveness of the proposed modification to the model structure presented in this study
derives from the work of Incerti et al. (2011) who found that a model with 3 EOM pools satisfactorily described the pattern
of litter decomposition. In addition, the authors found an enhancement of the predictive ability of the 3 pools model by
varying the decomposition rate of the pool with intermediate degradability as a function of the lignin content. Accordingly,
several authors have demonstrated that C simulation in amended soils is increased by a specific EOM parameterization
(Cavalli and Bechini, 2012; Peltre et al., 2012; Petersen et al., 2005b). The higher values of $K_{DEOM}$ and $K_{REOM}$ with respect
to $K_{DPM}$ and $K_{RPM}$ found in the present work underline the relevance to assess different decomposition rates for SOM and





EOM pools, in agreement with Cavalli et al. (2014) and Thuries et al. (2001), who found that EOM degradable and resistant pools always decomposed more rapidly than the analogue SOM pools. On a whole, all these results point out to the fact that existing models need to be adapted and calibrated in order to improve their ability to reliably simulate SOC trends in amended soils (Plaza et al., 2012).

## 4.3 Model optimization

In a preliminary test, the possibility to optimize the model following the procedure performed by Peltre et al. (2012), i.e. changing only the partition coefficients of DPM entry pools and utilizing the model standard decomposition rates, was investigated. Results showed that it was not possible to achieve a satisfactory fitting of the respiratory curve by only varying the proportion of DPM, RPM and HUM. As an example RMSE for soils amended with vine shoots compost, household waste compost and bioethanol residues was 17.9%, 20.0% and 29.9%, respectively. The optimization procedure was therefore carried out with the modified model by simultaneously varying the partition coefficients for DEOM, REOM and HEOM and the decomposition rate constants for DEOM and REOM. The results of the respiration curve fitting show that the modified model was able to adequately fit the respiratory response of amended soil, as demonstrated by the average value of RMSE for vine shoots compost (4.3%), household waste compost (2.5%), bioethanol residue (3.7%) and for all the 224 respiratory curve examined in this study (4.5%). As a comparison, Cavalli and Bechini (2011) calibrated the 3-EOM pools of CNSIM model for a reduced range of incubation conditions (3 soils and 5 liquid dairy manures) and obtained an average RMSE of 8.7%. Results suggest that for a reliable simulation of C mineralization in amended soils under laboratory conditions there is not only the need to partition EOM into a number of discrete pools, but also to find specific decomposition rates for such pools.

The calibration of EOM parameters was soil and incubation condition-specific to enable the model to find the best fit of measured data. Consequently, failures to simulate C trends can be attributed exclusively to the inadequacy of the model structure to accurately describe soil respiration. The results of the optimization procedure indicated that the modified model, encompassing additional EOM pools with specific parameters, is able to accommodate the large variability of the tested EOMs in terms of composition and properties. Such variability in EOM quality is indicated by the extended range of values characterizing each pool parameter. These findings support the hypothesis that explicit treatment of EOM heterogeneity would improve the performances of the RothC model. The lack of correlation between chemical properties of residues and pool parameters is in agreement to the evidence that operationally defined fractions do not precisely match kinetically defined pools. This is mainly due to the fact that the distinct components of organic residues interact with soil components and this modifies their decomposability along the incubation period (Trinsoutrout et al., 2000). Kinetically defined pools take into account such interactions and this represents an advantage in terms of simulation accuracy with respect to the operationally defined pools. The significant relationship between cumulative $CO_2$ and $f_{DMP}$ could be explained on the basis that most of the mineralized EOM-C emitted during the incubation derives from the degradable pool.





Calculation of mean pool parameters for EOM type and EOM group (Tables 6 and 7) indicated that the uncertainty associated to the parameters was always lower than the suggested threshold for statistically acceptable estimation of parameter (standard error of the mean < 50%; Robinson, 1985). These results indicate that the parameter values mainly reflects the EOM properties and that the model is capable to keep the effects of incubation conditions (i.e. type of soil, temperature, soil water content, rate of EOM application) to a minimum. This is in agreement with previous works

suggesting that EOM quality is the most important factor affecting organic residue decomposition in soil (Cavalli et al., 2014; Do Nascimento et al., 2012; Karhu et al., 2012;). A low variability associated to mean parameters for EOM group is an indication that this common set of parameters could be utilized to simulate SOC patterns in soil amended with the different EOMs belonging to a specific group with an acceptable error.

### 4.4 Potential limitations of the proposed model modification and optimization

Soil organic C modelling is subject to several potential drawbacks and limitations. Due to the aim of this work, only aspects specifically related to the proposed procedure for model modification and optimization will be discussed, namely the suitability of short term incubation to asses EOM pool parameters, issues of model validation for long term scale in field conditions and problems associated with simultaneous fitting of multiples parameters.

### 4.4.1 Suitability of short term incubations to asses EOM pool parameters and model validation under field conditions

One of the major concerns about the proposed optimization method is the suitability of short term incubation to adequately characterize EOM in terms of pools of different decomposability, as it was suggested that short term incubation are appropriate only to estimate the mineralization of the more decomposable pools. On the other hand, long term incubations, while providing a more accurate characterization of EOM, are highly demanding in terms of laboratory work, time and space. To date, an agreed minimum incubation period to obtain reliable evaluations of EOM pools has not been established.

Sleutel et al. (2005) underlined that such period depends on the EOM type and the kind of model used to fit the data. They found that for a specific EOM a reliable estimation was obtained with only 16 days at 16 °C and that a second order model required a minimum incubation time of about 50 days at 16 °C for the estimation of EOM stable organic C within less than 3% of the true value for all organic materials. In the case of a parallel first order model, a minimum incubation period of 42 days at 16 °C was necessary to obtain a reliable estimation of pig slurry and compost pools. Such EOMs represent well

stabilized materials for which the incubation time is likely to be more important for a reliable parameter estimation with respect to more degradable EOMs. Such minimum incubation periods are consistent with the incubation time utilized for most of the experiments in this study, when considering the different incubation temperature. It is important to note that one possible shortcut to reduce the incubation time needed for a satisfactory fitting is the use of higher temperatures, as the mineralization rates significantly increase. As a matter of fact, according to the rate modifying factor for temperature utilized

in RothC, an incubation period of 30 days at 20 °C corresponds to a period of 42 days at 16 °C to mineralize an equal amount of $CO_2$.



To verify the suitability of the incubation period utilized in this work (30 days) for a satisfactory curve fitting and test the dependence of EOM pool parameters from the incubation time, we have utilized an independent data set from a laboratory incubation performed at 15 and 25 °C for 300 days with a corn and soybean residues amended soil. We calibrated the EOM

pool parameters considering the whole incubation period (300 days) and a shorter incubation time (30 days for incubations at 25 °C and 50 days for incubations at 15° C). Results showed that the standard error associated to parameters obtained at different incubation time was acceptable (smaller than 50% of parameter value; Robinson, 1985). To estimate the error in SOC prediction associated with set of parameters calibrated at different incubation time, we performed long term RothC simulations (100 years), involving annual addition of 1 t ha$^{-1}$ of EOM-C and utilizing such different set of parameters.

Results showed that the difference in the yearly rate of SOC sequestration was always lower than 7%. We obtained similar results utilizing another set of independent data from a 60 days incubation of soil amended with cow manure, pig slurry and anaerobic digestates. In this case the difference in C sequestration potential utilizing set of parameters obtained after 30 and 60 days of incubation was lower than 5%.

The ability to estimate reliable SOM pools utilizing short term incubation data also depends on the accuracy in tracking the

cumulative respiratory curve. The high measurement frequency of the automatic system used in this study (1 measurement every 6 hours) improves the precision of the cumulative curve in comparison to standard methodologies (i.e. alkali trapping) characterized by a limited number of sampling points. High number of measurements allows outliers to be more easily identified and eliminated. A further source of uncertainty is related to the fact that cumulative curves accumulate errors associated with each sampling point. The system used in this study is characterized by a high precision, as percent relative

standard deviation of the mean for $CO_2$ measurement is typically less than 0.5% (Mondini et al., 2010). This minimizes the weight that each sampling point has on the total cumulative respiratory response in comparison to traditional measurements with alkali trapping, usually taken at large sampling intervals. We consider that the accuracy of measurement system utilized in this work compensate the possible limitations associated with short incubation time in comparison incubation performed for longer periods, but with less accurate and frequent measurements.

The reliability of short incubations in performing acceptable characterization of EOMs pool parameters is also supported by several researches. Mueller et al. (2003) used an incubation time of 52 days at 9 °C to calibrate EOM pools of DAISY model. It is important to note that an incubation period of 30 days at 20 °C, as the one carried out in our work, would correspond to an incubation period of 86 days at 9 °C to mineralize the same amount of $CO_2$. De Neve et al. (2003) incubated several wastes for 39 days at 21 °C to estimate the amount of stable C, a parameter that can be used directly as an

input is some C sequestration simulations models. Gale et al. (2006) showed that an incubation period of 28 days at 22 °C was sufficient for determining rate decomposition constants to represent EOM decomposition kinetics to be used in C models. Peltre et al. (2013) calibrated EOM pool parameter of the DAISY model utilizing respiration curves from amended soil incubated at 15 °C for 56 days. Saviozzi et al. (2014) performed an incubation of 25 days at 25 °C to infer the labile and recalcitrant EOM-C pool parameters. Similar conclusions concerning the suitability of short term incubation to obtain





reliable EOM characterization were drawn by Beloso et al. (1993), Pedra et al., (2007) and Garcia et al., (1992), utilizing incubation periods of 21, 28 and 42 days, respectively.

As a whole, an incubation time of 30 days and measurements performed with an accurate system could be considered as a reasonable trade off between the accuracy of the information obtained in terms of C mineralization and the demand for saving costs, time and space in the laboratory.

The suitability of short term incubation to estimate reliable EOM pools does not implies that parameters derived from short-term laboratory incubations can be automatically transposed to field condition to simulate long term C dynamics of amended soils. In fact, laboratory incubations are usually performed with sieved soil under optimal constant conditions for microbial activity that could result in quite different EOM mineralization rate with respect to that of a structured soil in a variable field environment. Therefore assimilation of laboratory data in existing models need to be carefully evaluated against field data

(Schimel et al., 2006). Nevertheless, several authors have demonstrated that model parameterization obtained in laboratory incubations can be utilized to provide reliable simulation of EOM mineralization under field conditions (Gabrielle et al., 2005; Kaborè et al., 2011; Noirot-Cosson et al., 2016; Vidal-Beaudet et al., 2012). This could be explained on the basis that EOMs composition and properties are the main factors regulating their mineralization in the soil (Cavalli et al., 2014; Do Nascimento et al., 2012; Karhu et al., 2012).

Furthermore, data requirements for model validation under field conditions makes at present this task not feasible to a large degree. As the main objective of the proposed model modification is to increase the ability to capture the large variability in EOM quality, validation at real scale would require data from long term field experiments dealing with a large range of EOM with contrasting properties. This is problematic to date due to the limited amount of suitable data available. While there are several ongoing long term experiments dealing with manure, straw and sludge amendment, there are relatively few

experiments reporting C data of soils amended with compost from source separate collection and anaerobic digestates. Furthermore, field experiments with EOM only recently utilized in agriculture (meat meals) or new EOMs from bioenergy by-products are lacking. Moreover, there are no long term field experiments dealing with EOMs characterized by different degree of stability (i.e. compost at different stages of the composting process) that would allow an improved validation of important model parameters such as pool partitioning factors ($f_{DEOM}$, $f_{REOM}$, $f_{HEOM}$) and $K_{REOM}$.

Another opportunity for enhanced validation would be the assessment of model ability to discriminate the effect of soil texture on EOM mineralization. Cavalli et al (2014), in an incubation study, reported no significant differences in EOM decomposition among soils with contrasting texture and attributed this to the soil structure disturbance caused by sample preparation in incubation experiments that can reduce the physical protection of the applied EOM and decrease a textural effect that might be more explicit under field conditions. Unfortunately experiments concerning EOM application to soils

with contrasting texture are very few.

Establishment of field experiments dealing with EOM characterized by contrasting properties and degree of transformation under different environmental and management options would allow for a proper evaluation of model performances in simulating long term SOC dynamics.



### 4.4.2 Simultaneous fitting of multiple parameters

In RothC, as in many other SOM models, the amount of C associated with each pool decomposes following an exponential decay. In theory these pools are of defined size that should not change with environmental conditions or with the procedure used to fit the model with the data. Cabrera et al. (1995) underlined that pools and rate constants in the exponential models are inversely related which suggests that the same fit to available data could be obtained by increasing one parameter while decreasing the other. Research has also shown that increasing the incubation time can increase or decrease the size of a pool

while having the opposite effect on rate constants. The possibility to obtain non unique set of parameters also increases with decreasing number of incubation data. These problems with exponential models suggest that they need to be used judiciously when trying to identify pools of defined and fixed size as different combinations of pool size and decomposition rate giving a good fit to respiratory curve may result in significant differences in SOC when the model is run over a long term period.

A possible solution to avoid this pitfall is to have independent controls to constrain parameters estimates (Ahrens et al.,

2014). Unfortunately, we did not have such controls for all the incubation data. Nevertheless, we are confident that the optimized parameters represent a univocal set of values. First of all, univocality in the optimized parameters was sought by maintaining constant HEOM decomposition rate and by imposing constraints to partition coefficients and decomposition rates according to scientific data in order to obtain pool parameters biologically meaningful. As for the influence of incubation time on pool estimates, we have found consistent set of parameters between calibrations performed after 30 and

300 days of incubation. Regarding the impact of few measurements points, this does not apply to our curves characterized by a high frequency measurement. Moreover, even if from a theoretical point of view there is the possibility to obtain different set of parameters leading to accurate simulation, this is limited by the shape of the cumulative curve, as for example the first part of the curve, describing the fast release of $CO_2$ from the most degradable C, can be adequately described only by a specific combination of $k_{DEOM}$ and $f_{DPEOM}$ values. Finally, a significant correlation between pool size and decomposition is an

indication of model overparameterization and the likelihood to obtain accurate simulations by different combination of the parameters. In the case of the present work such relationships were always not significant and this suggests that the optimized parameters are likely to reflect a unique solution. Simultaneous fitting of several parameters is not unusual in model calibration. As an example, Muller et al. (2003) and Cavalli and Bechini (2012) simultaneously fitted 5 and 6 parameters, respectively.

**5 Conclusions**

An effective management of organic amendment requires the development of C models able to take into account the quality of added EOM. The main innovative aspects of this work consist in the modification of the RothC model to include additional EOM pools and in their parameterization by model fitting to respiratory curves of amended soils. Results of the study show that the modified and optimized model was able to adequately describe EOM mineralization curves obtained



under laboratory conditions and support the hypothesis that defining EOM-specific partitioning factors and decomposition rates improves the simulation ability of the model in amended soils.

Due to the effect of different environmental conditions between laboratory and field conditions, the validation of the modified model with field data represents a necessary step in the model development as a tool to evaluate SOC storage in EOM amended soils in the long term. However, the conceptual changes to the model structure and the potential usefulness of

the model are justified through its ability to simulate detailed experimental data. We consider that the capacity of the model to adequately describe mineralization curves of EOM under laboratory conditions represents an essential prerequisite for a reliable C modelling of amended soils, as it demonstrates the ability of the model to resolve the large variability in EOM composition and properties. Furthermore, information derived from the fitting procedure could be useful to identify knowledge gaps on environmental factors and soil processes that regulate EOM decomposition in the soil, suggesting further

ways to improve the model.

The findings of the present research indicate that laboratory experiments on EOM decomposition could be useful to improve the simulation of C dynamics in amended soils.

**Authors contribution**

Claudio Mondini conceived and designed the experiments, analyzed the data, wrote the paper, prepared figures and/or tables

and reviewed drafts of the paper.

Maria Luz Cayuela conceived, designed and performed the experiments, analyzed the data, and reviewed drafts of the paper.

Tania Sinicco performed the experiments and reviewed drafts of the paper.

Flavio Fornasier conceived and designed the experiments, performed the experiments and reviewed drafts of the paper.

Antonia Galvez performed the experiments and reviewed drafts of the paper.

Miguel Angel Sánchez-Monedero conceived, designed and performed the experiments and reviewed drafts of the paper.

**Acknowledgments**

The authors would like to express their gratitude to Emanuela Vida for keen dedication and skillful work in performing the incubations experiments and the laboratory analysis. The authors are very deeply grateful to Kevin Coleman for information and explanations on the RothC model and for advice and suggestions on the modification of the model.

This study was performed under the framework of the EU project FP7 KBBE.2011.1.2-02 FERTIPLUS "Reducing mineral fertilisers and agro-chemicals by recycling treated organic waste as compost and bio-char" project nr. 289853 co-funded by the European Com-mission, Directorate General for Research & Innovation, within the 7[th] Framework Programme of RTD, Theme 2 – Biotechnologies, Agriculture & Food. This publication reflects the author's views, findings and conclusions and



the European Commission is not liable for any use that may be made of the information contained therein. M.L. Cayuela is
supported by a "Ramón y Cajal" research contract from the Spanish Ministry of Economy and Competitiveness.

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





**Table 1.** Soil C models utilized for C simulation in amended soils.

| Model | EOMs | Yearly application rate | Simulation period (y) | Reference |
|---|---|---|---|---|
| RothC | Chicken and dairy manure | 170-670 kg N ha$^{-1}$ | 2 | Abbas and Fares, 2009 |
| RothC | Cattle and pig FYM and slurry, broiler litter | 0.6-7.0 t C ha$^{-1}$ | 14 | Bhogal et al., 2010 |
| RothC | FYM, WS, SS, sawdust, compost | 6.5-30 t ha$^{-1}$ | 11-52 | Peltre et al., 2012 |
| RothC | User defined | User defined | User defined | Carbo-PRO web tool, 2012 |
| RothC | FYM | 10-15 t fw ha$^{-1}$ | 25 | Yokozawa et al., 2010 |
| RothC | Waste garden compost | 5-45 t ha$^{-1}$ | 15 | Tits et al., 2014 |
| C-simulator | Waste garden and household waste compost | 30 t ha$^{-1}$ | 13 | Tits et al., 2010 |
| CN-SIM | FYM | 2 t C ha$^{-1}$ | 52 | Petersen et al., 2005a |
| DAISY | Compost | 5-10 t dm ha$^{-1}$ | 50 | Stoppler-Zimmer et al., 1999 |
| DAISY | Oilseed rape straw | 8 t ha$^{-1}$ | 2 | Mueller et al., 1997 |
| DAISY | WS, maize, blue grass | 6 t fw ha$^{-1}$ | 1 | Mueller et al., 1998 |
| DAISY | FYM, WS, sawdust | 6.5 t dm ha$^{-1}$ | 35 | Bruun et al., 2003 |
| DAISY | MSW compost, SS, FYM, cattle slurry | 200 kg N ha$^{-1}$ | 50 | Peltre et al., 2013 |
| DAISY | Compost | 20 t ha$^{-1}$ | 4.5 | Gerke et al., 1999 |
| NCSOIL | MSW compost | 10-25 t dm ha$^{-1}$ | 4 | Gabrielle et al., 2005 |
| NCSOIL | FYM, Urban waste compost | 2 t C ha$^{-1}$ | 13 | Noirot-Cosson et al., 2016 |
| Cantis | WS | 8 t dm ha$^{-1}$ (1.2 g C kg$^{-1}$) | 1 | Garnier et al., 2003 |
| Yasso07 | WS, FYM, green manure | 2 t C ha$^{-1}$ | 35 | Karhu et al., 2012 |
| DNDC | WS, FYM, compost | 0.03-0.5 t C ha$^{-1}$ | 6 | Sleutel et al., 2006 |
| CENTURY | WS, FYM, sawdust, green manure | 2 t C ha$^{-1}$ | 30 | Paustian et al., 1992 |
| CQESTR | WS, FYM, corn stalks | 6.0-7.5 t dm ha$^{-1}$ | 34 | Plaza et al., 2012 |
| 3 pools model | Organic compost | 20-40% w:w | 5 | Vidal-Beaudet et al., 2012 |

EOM: exogenous organic matter; FYM: farmyard manure; WS: wheat straw; SS: sewage sludge; MSW: municipal solid wastes; fw: fresh weight; dm: dry matter; w: weight.



**Table 2.** Main physico-chemical characteristics of soils used for incubations.

| Location | Country | Soil code | Soil use | Sand (%) | Silt (%) | Clay (%) | pH | CaCO$_3$ (g kg$^{-1}$) | SOC (g kg$^{-1}$) | N$_{TOT}$ (g kg$^{-1}$) | SOC/N$_{TOT}$ | C$_{mic}$ (µg g$^{-1}$) |
|---|---|---|---|---|---|---|---|---|---|---|---|---|
| S. Martino | Italy | SM | Arable | 69 | 28 | 3 | 8.3 | 740 | 10.5 | 1.2 | 8.8 | 114 |
| Gorizia | Italy | GO | Meadow | 37 | 48 | 15 | 7.8 | 46 | 25.4 | 2.4 | 10.6 | 795 |
| Bueris | Italy | BU | Arable | 6.0 | 48 | 46 | 7.0 | - | 32.0 | 4.5 | 7.1 | 269 |
| Lodi | Italy | LO | Meadow | 67 | 21 | 12 | 6.7 | - | 22.0 | 2.1 | 10.5 | 205 |
| Reana | Italy | PE | Arable | 55 | 28 | 17 | 6.5 | - | 15.9 | 1.2 | 13.3 | 118 |
| Ribis | Italy | RI | Arable | 54 | 32 | 14 | 4.6 | - | 8.1 | 1.3 | 6.2 | 65 |
| Codroipo | Italy | CO | Arable | 27 | 58 | 15 | 7.1 | - | 19.0 | 2.0 | 9.5 | 350 |
| Jumilla | Spain | JU | Olive orchard | 52 | 21 | 27 | 8.0 | 415 | 10.4 | 1.0 | 10.4 | 119 |
| Alquife | Spain | AL | Disused mine | 53 | 30 | 17 | 8.5 | 1.3 | 2.5 | 0.9 | 2.8 | 10 |
| Llano de la Perdiz | Spain | LL | Arable | 32 | 17 | 51 | 7.0 | 0.5 | 9.2 | 1.1 | 8.4 | 146 |

SOC: soil organic C; N$_{TOT}$: total N; C$_{mic}$: soil microbial biomass C.





**Table 3.** Description of exogenous organic matter (EOM) used for incubations.

| EOM group | EOM group code | EOM type | EOM type description | EOM type code |
|---|---|---|---|---|
| Compost | CO | Vine shoots compost | compost from vine tree prunings | VSC |
| | | Household waste compost | compost from the separate collection of household organic wastes | HWC |
| | | Green waste compost | compost from green waste | GWC |
| | | CC + WS + MM II_3 | 3 days old compost from a mixture of cotton cardings, wheat straw and meat and bone meal | CMC II |
| | | CC + WS + MM III_9 | 9 days old compost from a mixture of cotton cardings, wheat straw and meat and bone meal | CMC III |
| | | CC + WS + MM M_92 | 92 days old compost from a mixture of cotton cardings, wheat straw and meat and bone meal | CMC M |
| | | CC + WS + BLM + HHM II_3 | 3 days old compost from a mixture of cotton cardings, wheat straw, blood meal and hoof and horn meal | CBC II |
| | | CC + WS + BLM + HHM III_9 | 9 days old compost from a mixture of cotton cardings, wheat straw, blood meal and hoof and horn meal | CBC III |
| | | CC + WS + BLM + HHM IV_21 | 21 days old compost from a mixture of cotton cardings, wheat straw, blood meal and hoof and horn meal | CBC IV |
| | | CC + WS + BLM + HHM M_92 | 92 days old compost from a mixture of cotton cardings, wheat straw, blood meal and hoof and horn meal | CBC M |
| Bioenergy by-products | BE | Bioethanol residue | wheat starch by-product from bioethanol production | BR |
| | | Rapeseed meal | meal from biodiesel production | RSM |
| Anaerobic digestates | AD | Pig slurry digestate | anaerobic digestate of pig slurry | PS |
| | | TPOMW + manure digestate | mesophilic anaerobic digestates of two-phase olive mill waste and liquid manure | OW 1 |
| | | TPOMW + manure digestate (55 °C) | thermophilic anaerobic digestates of two-phase olive mill waste and liquid manure | OW 2 |
| | | Liquid manure digestate | anaerobic digestate of liquid manure | OW 3 |
| | | TPOMW digestate | anaerobic digestate of two-phase olive mill waste | OW 4 |
| Meat and bone meals | MM | Bovine MM 1 | bovine meat and bone meal | BV1 |
| | | Bovine MM 2 | bovine meat and bone meal | BV2 |
| | | Swine MM | swine meat and bone meal | SW |
| | | Mixed swine bovine MM | mixture of swine and bovine meat and bone meal | SB |
| | | Defatted bovine MM | defatted bovine meat and bone meal | DE |
| Animal residues | AR | Hydrolyzed leather | organic fertiliser derived from hydrolysed animal proteins | HL |
| | | Blood meal | organic fertiliser from spray drying at low temperatures fresh whole blood from animal processing plants | BLM |
| | | Horn and Hoof meal | organic fertiliser produced by the drying of horns and hooves from animal processing plants | HHM |
| Crop residues | CR | Cotton cardings | waste derived from the process of preparing the fibers of cotton (Gossypium spp., L.) for spinning | CC |
| | | Wheat straw | winter wheat (Triticum aestivum L.) straw collected after harvesting | WS |
| Agro-industrial wastes | AW | Two-phase olive mill waste | semisolid sludge generated during the extraction of olive oil by the two-phase centrifugation system | TPOMW |
| Sewage sludges | SS | Wastewater sludge | sewage sludge from an urban wastewater treatment plant | WW |
| Biochars | BC | Green waste biochar | biochar produced by continuous slow pyrolysis of green waste at 550 °C | GWB |

CC: cotton cardings; WS: wheat straw; MM: meat and bone meal; BLM: blood meal; HHM: hoof and horn meal; TPMOW: two-phase olive mill waste. Compost: roman numerals refer
to stages of the process, numbers refer to days of composting, M: mature compost.



**Table 4.** Main chemical characteristics of exogenous organic matter (EOM) used for incubations.

| EOM group | EOM group code | EOM type | EOM type code | pH | OM (%) | TOC (%) | $N_{TOT}$ (%) | $TOC/N_{TOT}$ | WSC (g kg⁻¹) | WSN (g kg⁻¹) | $NH_4^+$ (mg kg⁻¹) | $NO_3^-$ (mg kg⁻¹) |
|---|---|---|---|---|---|---|---|---|---|---|---|---|
| Compost | CO | Vine shoots compost | VSC | 7.7 | 65.4 | 34.5 | 1.5 | 23.2 | 9.8 | 0.6 | 101 | 2018 |
| | | Household waste compost | HWC | 8.3 | 64.2 | 34.4 | 2.3 | 14.9 | 5.9 | 0.7 | 226 | 1777 |
| | | Green waste compost | GWC | 7.5 | 51.9 | 28.2 | 2.2 | 12.8 | 6.8 | 2.2 | 1400 | 800 |
| | | CC + WS + MM II_3 | CMC II | 7.9 | 90.6 | 42.7 | 1.6 | 27.2 | 23.1 | 2.7 | 627 | 2397 |
| | | CC + WS + MM III_9 | CMC III | 7.9 | 89.3 | 41.3 | 2.1 | 19.6 | 28.3 | 3.7 | 96 | 2821 |
| | | CC + WS + MM M_92 | CMC M | 8.1 | 82.2 | 39.9 | 3.5 | 11.3 | 12.4 | 1.8 | 254 | 2195 |
| | | CC + WS + BLM + HHM II_3 | CBC II | 7.7 | 93.3 | 43.5 | 1.7 | 25.7 | 20.3 | 1.9 | 434 | 1901 |
| | | CC + WS + BLM + HHM III_9 | CBC III | 7.9 | 91.2 | 42.9 | 1.9 | 22.0 | 21.9 | 2.4 | 281 | 2256 |
| | | CC + WS + BLM + HHM IV_21 | CBC IV | 7.9 | 87.5 | 42.3 | 2.8 | 15.2 | 23.6 | 2.5 | 134 | 2155 |
| | | CC + WS + BLM + HHM M_92 | CBC M | 7.7 | 83.2 | 40.6 | 3.7 | 10.9 | 11.3 | 3.4 | 59 | 4169 |
| Bioenergy by-products | BE | Bioethanol residue | BR | 4.2 | 91.9 | 48.5 | 6.2 | 7.8 | 202.5 | 12.8 | 1153 | 1.9 |
| | | Rapeseed meal | RSM | 6.2 | 92.5 | 45.9 | 6.0 | 7.7 | 74.4 | 2.4 | 180 | 13.4 |
| Anaerobic digestates | AD | Pig slurry digestate | PS | 8.5 | 74.6 | 37.9 | 4.4 | 8.7 | 38.7 | 11.6 | 5361 | 11048 |
| | | TPOMW + manure digestate | OW 1 | 8.2 | 75.8 | 43.6 | 3.4 | 13.0 | 23.1 | 1.78 | 17322 | 14318 |
| | | TPOMW + manure (55 °C) digestate | OW 2 | 8.1 | 76.7 | 48.7 | 3.5 | 13.9 | 25.1 | 1.81 | 20118 | 7902 |
| | | Liquid manure digestate | OW 3 | 8.2 | 68.6 | 44.1 | 3.1 | 14.5 | 13.6 | 0.94 | 690 | 10905 |
| | | TPOMW digestate | OW 4 | 7.9 | 76.9 | 49.1 | 3.0 | 16.1 | 17.7 | 1.07 | 18744 | 10905 |
| Meat and bone meals | MM | Bovine MM 1 | BV1 | 6.5 | 70.9 | 38.5 | 8.4 | 4.6 | 52.1 | 12.4 | 631 | 1692 |
| | | Bovine MM 2 | BV2 | 6.3 | 65.5 | 33.9 | 8.2 | 4.1 | 35.7 | 9.0 | 410 | 1314 |
| | | Swine MM | SW | 6.7 | 78.6 | 41.4 | 9.0 | 4.6 | 114.5 | 29.0 | 530 | 4721 |
| | | Mixed swine bovine MM | SB | 5.9 | 81.8 | 43.1 | 9.4 | 4.6 | 60.5 | 17.1 | 359 | 4165 |
| | | Defatted bovine MM | DE | 6.4 | 59.4 | 29.9 | 8.4 | 3.6 | 34.3 | 9.9 | 435 | 1714 |
| Animal residues | AR | Hydrolyzed leather | HL | 5.2 | 80.0 | 42.0 | 13.2 | 3.2 | 34.7 | 23.2 | 6360 | 3135 |
| | | Blood meal | BLM | 6.7 | 90.8 | 52.6 | 16.4 | 3.2 | 118.9 | 37.9 | 122 | 3547 |
| | | Horn and Hoof meal | HHM | 7.5 | 80.0 | 51.3 | 17.0 | 3.0 | 13.7 | 5.0 | 1887 | 1058 |
| Crop residues | CR | Cotton cardings | CC | 6.2 | 88.0 | 45.2 | 1.5 | 30.5 | 37.8 | 2.4 | 303 | 2005 |
| | | Wheat straw | WS | 6.5 | 89.1 | 49.6 | 0.3 | 198 | 15.7 | 0.8 | 66 | 879 |
| Agro-industrial wastes | AW | Two-phase olive mill waste | TPOMW | 5.3 | 94.1 | 53.0 | 1.3 | 41.1 | 4.1 | 1.1 | 122 | 0 |
| Sewage sludges | SS | Wastewater sludge | WW | 6.8 | 70.9 | 38.4 | 4.8 | 8.0 | 7.97 | 1.64 | 1677 | 2286 |
| Biochars | BC | Green waste biochar | GWB | 7.5 | 98.3 | 86.3 | 0.3 | 345 | 0.1 | 0.0 | 17 | 0.4 |
| | | | *Mean* | *7.1* | *80.1* | *43.8* | *5.0* | *30.6* | *36.3* | *6.8* | *2670* | *3107* |
| | | | *Minimum* | *4.2* | *51.9* | *28.2* | *0.3* | *3.0* | *0.1* | *0.0* | *17* | *0.0* |
| | | | *Maximum* | *8.5* | *98.3* | *86.3* | *17.0* | *345* | *203* | *37.9* | *20118* | *14318* |

EOM: exogenous organic matter; OM: organic matter; TOC: total organic C; $N_{TOT}$: total N; WSC: water soluble C; WSN: water soluble N CC: cotton cardings; WS: wheat straw; MM:

meat and bone meal; BLM: blood meal; HHM: hoof and horn meal; TPMOW: two-phase olive mill waste.

Compost: roman numerals refer to stages of the process, numbers refer to days of composting, M: mature compost.

For EOM group and type code refer to Table 3.



**Table 5**. Cumulative extra $CO_2$-C emitted in amended soil (% of added C) for exogenous organic matter (EOM) group for all

incubations and incubations performed under standard conditions.

| EOM group | All incubations | | | | Standard conditions incubations* | | | |
|---|---|---|---|---|---|---|---|---|
| | Mean | Min | Max | n | Mean | Min | Max | n |
| | $CO_2$-C (%) | | | | $CO_2$-C (%) | | | |
| Biochar | 0.02 | 0.01 | 0.04 | 4 | 0.02 | 0.01 | 0.04 | 4 |
| Compost | 3.7 | 0.9 | 11.1 | 34 | 3.0 | 0.9 | 6.6 | 19 |
| Bioenergy by-products | 12.9 | 6.9 | 16.8 | 20 | 12.8 | 6.9 | 16.8 | 10 |
| Anaerobic digestates | 3.8 | 0.8 | 7.2 | 27 | 4.0 | 0.8 | 7.1 | 10 |
| Meat and bone meals | 16.8 | 7.8 | 38.6 | 93 | 21.3 | 18.1 | 25.9 | 3 |
| Animal residues | 13.1 | 5.0 | 21.1 | 33 | 16.8 | 11.0 | 21.1 | 14 |
| Crop residues | 8.5 | 3.0 | 18.4 | 10 | 10.4 | 5.1 | 18.4 | 6 |
| Agro-industrial wastes | 10.0 | 6.0 | 17.5 | 3 | 6.3 | 6.0 | 17.5 | 3 |
| Sewage sludges | 4.8 | 3.8 | 6.0 | 4 | 4.8 | 3.8 | 6.0 | 4 |
| **Total cases** | | | | **228** | | | | **69** |

* 20 °C, 40 % soil water holding capacity, 0.5 % EOM application rate and 30 days incubation period.





**Table 6**. Mean, minimum and maximum values of statistical indicators of model goodness of the fit between measured and

simulated data (n = 224).

| | RMSE | R | E | M |
|---|---|---|---|---|
| | **%** | | **%** | **µg $CO_2$-C $g^{-1}$** |
| **Mean** | 4.5 | 0.995 | -1.1 | -1.2 |
| **Min** | 0.7 | 0.794 | -16.4 | -44.8 |
| **Max** | 37.2 | 0.9999 | 3.5 | 2.9 |

RMSE: root mean square error; R: sample correlation coefficient; E: relative error M: mean difference between measured

and simulated data.



**Table 7.** Mean RothC exogenous organic matter (EOM) pool parameters for different EOM types.

| EOM group | EOM type code | N. | Exc. | Inc. | Mean value $f_{DEOM}$ | $f_{REOM}$ | $f_{HEOM}$ | $K_{DEOM}$ | $K_{REOM}$ | Standard error (SE) $f_{DEOM}$ | $f_{REOM}$ | $f_{HEOM}$ | $K_{DEOM}$ | $K_{REOM}$ | Coeff. of variation of SE (%) $f_{DEOM}$ | $f_{REOM}$ | $f_{HEOM}$ | $K_{DEOM}$ | $K_{REOM}$ |
|---|---|---|---|---|---|---|---|---|---|---|---|---|---|---|---|---|---|---|---|
|  | VSC | 4 | 0 | 4 | 0.01 | 0.39 | 0.59 | 119 | 0.23 | 0.005 | 0.032 | 0.03 | 42.09 | 0.04 | 41.1 | 8.1 | 5.7 | 35.3 | 19.2 |
|  | HWC | 14 | 0 | 14 | 0.02 | 0.33 | 0.65 | 78 | 0.28 | 0.002 | 0.018 | 0.02 | 13.46 | 0.05 | 8.5 | 5.3 | 2.7 | 17.3 | 18.8 |
|  | GWC | 2 | 0 | 2 | 0.01 | 0.31 | 0.69 | 145 | 0.45 | 0.001 | 0.043 | 0.04 | 45.50 | 0.07 | 10.0 | 14.0 | 6.6 | 31.5 | 15.9 |
|  | CMC II | 2 | 0 | 2 | 0.05 | 0.87 | 0.08 | 42 | 0.34 | 0.009 | 0.025 | 0.02 | 3.18 | 0.03 | 16.6 | 2.9 | 20.3 | 7.6 | 8.2 |
|  | CMC III | 2 | 0 | 2 | 0.06 | 0.63 | 0.32 | 29 | 0.35 | 0.005 | 0.006 | 0.00 | 1.20 | 0.02 | 8.2 | 1.0 | 0.5 | 4.1 | 6.6 |
| Compost | CMC M | 2 | 0 | 2 | 0.004 | 0.26 | 0.74 | 83 | 0.46 | 0.002 | 0.047 | 0.04 | 21.77 | 0.11 | 58.3 | 18.1 | 6.0 | 26.1 | 22.8 |
|  | CBC II | 2 | 0 | 2 | 0.08 | 0.73 | 0.19 | 22 | 0.28 | 0.014 | 0.015 | 0.00 | 0.24 | 0.05 | 17.4 | 2.1 | 0.6 | 1.1 | 17.2 |
|  | CBC III | 2 | 0 | 2 | 0.07 | 0.62 | 0.31 | 20 | 0.26 | 0.001 | 0.031 | 0.03 | 1.47 | 0.10 | 2.0 | 5.0 | 9.8 | 7.4 | 39.3 |
|  | CBC IV | 2 | 0 | 2 | 0.05 | 0.56 | 0.39 | 16 | 0.22 | 0.001 | 0.012 | 0.01 | 0.66 | 0.01 | 2.2 | 2.2 | 2.6 | 4.0 | 5.1 |
|  | CBC M | 2 | 0 | 2 | 0.01 | 0.31 | 0.69 | 145 | 0.45 | 0.001 | 0.043 | 0.04 | 45.50 | 0.07 | 10.0 | 14.0 | 6.6 | 31.5 | 15.9 |
| Bioenergy by products | BR | 6 | 1 | 5 | 0.12 | 0.88 |  | 129 | 0.47 | 0.007 | 0.007 |  | 7.36 | 0.07 | 5.8 | 0.8 |  | 5.7 | 14.4 |
|  | RSM | 14 | 2 | 12 | 0.13 | 0.87 |  | 76 | 0.27 | 0.007 | 0.007 |  | 6.51 | 0.03 | 5.5 | 0.8 |  | 8.5 | 9.6 |
| Anaerobic digestates | PS | 14 | 2 | 12 | 0.05 | 0.70 | 0.25 | 57 | 0.25 | 0.004 | 0.024 | 0.02 | 4.92 | 0.03 | 7.4 | 3.5 | 9.1 | 8.6 | 12.9 |
|  | OW | 13 | 0 | 13 | 0.01 | 0.74 | 0.25 | 220 | 0.20 | 0.002 | 0.018 | 0.02 | 23.34 | 0.03 | 13.7 | 2.5 | 7.2 | 10.6 | 13.5 |
|  | BV1 | 26 | 0 | 26 | 0.19 | 0.81 |  | 78 | 0.47 | 0.011 | 0.011 |  | 2.68 | 0.08 | 5.5 | 1.3 |  | 3.4 | 16.7 |
| Meat and bone meals | SB | 16 | 0 | 16 | 0.29 | 0.71 |  | 56 | 0.33 | 0.043 | 0.043 |  | 5.51 | 0.06 | 14.8 | 6.0 |  | 9.8 | 17.9 |
|  | BV2 | 40 | 4 | 36 | 0.16 | 0.84 |  | 81 | 0.29 | 0.005 | 0.005 |  | 2.24 | 0.03 | 3.3 | 0.6 |  | 2.8 | 10.3 |
|  | DE | 10 | 0 | 10 | 0.32 | 0.68 |  | 59 | 0.39 | 0.031 | 0.031 |  | 5.29 | 0.12 | 9 | 5 |  | 9 | 31.5 |
|  | HLM | 3 | 0 | 3 | 0.15 | 0.85 |  | 67 | 0.67 | 0.030 | 0.030 |  | 19.82 | 0.22 | 19.7 | 3.6 |  | 29.4 | 32.5 |
| Animal residues | BLM | 15 | 1 | 14 | 0.10 | 0.90 |  | 164 | 0.40 | 0.012 | 0.012 |  | 14.27 | 0.07 | 11.2 | 1.3 |  | 8.7 | 17.0 |
|  | BLM2 | 3 | 0 | 3 | 0.13 | 0.87 |  | 217 | 0.90 | 0.034 | 0.034 |  | 50.12 | 0.16 | 25.3 | 3.9 |  | 23.1 | 17.3 |
|  | HHM | 12 | 2 | 10 | 0.23 | 0.77 |  | 16 | 0.19 | 0.027 | 0.027 |  | 1.42 | 0.02 | 11.7 | 3.6 |  | 8.6 | 9.6 |
| Vegetal residues | CC | 5 | 1 | 4 | 0.05 | 0.95 |  | 87 | 0.35 | 0.012 | 0.012 |  | 24.14 | 0.11 | 25.0 | 1.2 |  | 27.9 | 30.1 |
|  | WS | 5 | 1 | 4 | 0.05 | 0.95 |  | 39 | 0.19 | 0.011 | 0.011 |  | 4.34 | 0.04 | 23.1 | 1.1 |  | 11.2 | 18.2 |
| Agro-industrial wastes | TPOMW | 3 | 1 | 2 | 0.04 | 0.78 | 0.19 | 126 | 0.56 | 0.011 | 0.011 | 0.0001 | 6.76 | 0.25 | 28.7 | 1.5 | 0.1 | 5.4 | 44.3 |
| Sewage sludge | WW | 4 | 0 | 4 | 0.04 | 0.96 |  | 63 | 0.22 | 0.002 | 0.002 |  | 7.75 | 0.04 | 4.7 | 0.2 |  | 12.3 | 18.5 |
| **Total (N.)** |  | 223 | 15.0 | 208 | 0.09 | 0.70 | 0.41 | 86 | 0.37 |  |  |  |  |  | mean 15.0 | 4.2 | 6.0 | 13.5 | 18.6 |
| **Total (%)** |  | 100 | 6.7 | 93.3 | 0.004 | 0.26 | 0.08 | 16 | 0.19 |  |  |  |  |  | minimum 2.0 | 0.2 | 0.1 | 1.1 | 5.1 |
|  |  |  |  |  | 0.32 | 0.96 | 0.74 | 220 | 0.90 |  |  |  |  |  | maximum 58.3 | 18.1 | 20.3 | 35.3 | 44.3 |

EOM: exogenous organic matter; N.: number of incubations; Exc./Inc.: number of incubations excluded/included from the mean calculation; DEOM: decomposable EOM; REOM: resistant EOM; HEOM: humified EOM; f: partitioning factor; K: decomposition constant rate (y$^{-1}$).

For EOM group and type code refer to Table 3.

**Table 8.** Mean RothC exogenous organic matter (EOM) pool parameters for different EOM groups.

| EOM group | EOM group code | N. | Excl. | Incl. | Mean value | | | | | Standard error (SE) | | | | | Coefficient of variation of SE (%) | | | | |
|---|---|---|---|---|---|---|---|---|---|---|---|---|---|---|---|---|---|---|---|
| | | | | | $f_{DEOM}$ | $f_{REOM}$ | $f_{HEOM}$ | $K_{DEOM}$ | $K_{REOM}$ | $f_{DEOM}$ | $f_{REOM}$ | $f_{HEOM}$ | $K_{DEOM}$ | $K_{REOM}$ | $f_{DEOM}$ | $f_{REOM}$ | $f_{HEOM}$ | $K_{DEOM}$ | $K_{REOM}$ |
| Compost | CO | 34 | 0 | 34 | 0.03 | 0.44 | 0.53 | 79 | 0.30 | 0.004 | 0.031 | 0.034 | 11 | 0.027 | 14.6 | 6.9 | 6.4 | 13.8 | 8.8 |
| Bioenergy by-products | BE | 20 | 3 | 17 | 0.13 | 0.87 | | 92 | 0.33 | 0.006 | 0.006 | | 8 | 0.035 | 4.3 | 0.6 | | 8.5 | 10.4 |
| Anaerobic digestates | AD | 27 | 2 | 25 | 0.03 | 0.74 | 0.25 | 220 | 0.20 | 0.004 | 0.018 | 0.018 | 23 | 0.027 | 14.9 | 2.5 | 7.2 | 10.6 | 13.5 |
| Meat and bone meals | MM | 93 | 4 | 89 | 0.21 | 0.79 | | 74 | 0.41 | 0.011 | 0.011 | | 2 | 0.039 | 5.1 | 1.4 | | 2.8 | 9.5 |
| Animal residues | AR | 33 | 3 | 30 | 0.15 | 0.85 | | 110 | 0.41 | 0.015 | 0.015 | | 16 | 0.056 | 10.0 | 1.8 | | 14.5 | 13.7 |
| Crop residues | CR | 10 | 2 | 8 | 0.05 | 0.95 | | 63 | 0.27 | 0.007 | 0.007 | | 15 | 0.060 | 15.7 | 0.8 | | 23.2 | 22.0 |
| Agro-industrial wastes | AW | 3 | 1 | 2 | 0.04 | 0.78 | 0.19 | 126 | 0.56 | 0.011 | 0.011 | 0.0001 | 7 | 0.249 | 28.7 | 1.5 | 0.1 | 5.4 | 44.3 |
| Sewage Sludges | SS | 4 | 0 | 4 | 0.04 | 0.96 | | 63 | 0.22 | 0.002 | 0.002 | | 8 | 0.040 | 4.7 | 0.2 | | 12.3 | 18.5 |
| **Total (N.)** | | 224 | 15 | 209 | | | | | | | | | | mean | 12.2 | 2.0 | 4.6 | 11.4 | 17.6 |
| **Total (%)** | | 100 | 6.7 | 93 | | | | | | | | | | minimum | 4.3 | 0.2 | 0.1 | 2.8 | 8.8 |
| | | | | | | | | | | | | | | maximum | 28.7 | 6.9 | 7.2 | 23.2 | 44.3 |

EOM: exogenous organic matter; N.: number of incubations; Exc./Inc.: number of incubations excluded/included from the mean calculation; DEOM: decomposable EOM; REOM:

resistant EOM; HEOM: humified EOM; f: partitioning factor; K: decomposition constant rate (y⁻¹).

For EOM code refer to Table 3.



**Figure captions**

Figure 1.    Diagram of the automated chromatographic system for soil $CO_2$ sampling and measurement.

Figure 2.    Structure of the standard (a) and modified (b) RothC model. DPM: decomposable plant material; RPM: resistant plant material; EOM: exogenous organic matter; DEOM: decomposable EOM; REOM: resistant EOM; HEOM: humified EOM; BIO: soil microbial biomass; HUM: humified soil organic matter; IOM: inert organic matter; f: partitioning factor; K: decomposition constant rate ($y^{-1}$).

Figure 3.    Rate (a) and net cumulative measured and simulated (b) $CO_2$ emission from Llano de la Perdiz soil amended with amendments of different degree of degradability during a 30 days laboratory incubation. For the rate of respiration only the first 10 days of the incubation are reported. Simulated net cumulative respiration curves are represented as dotted lines. Respiratory curves are presented on Y axis with different scale for better visualization.





Figure 1

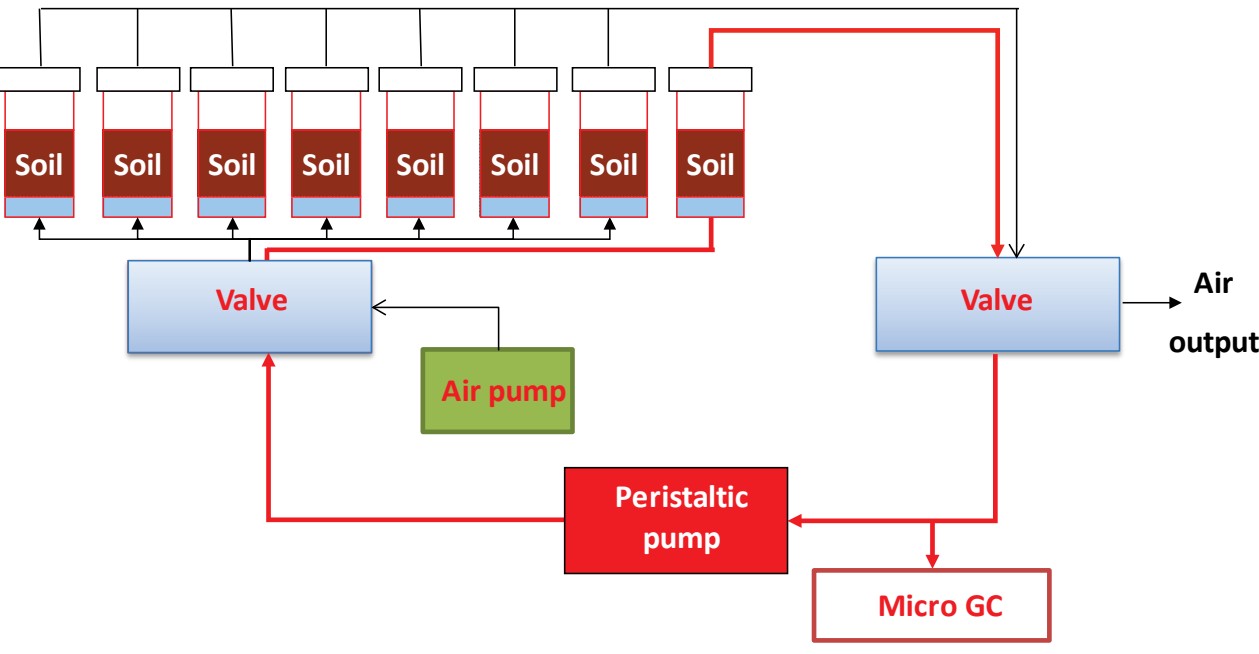

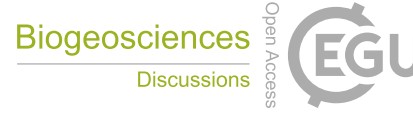

Figure 2

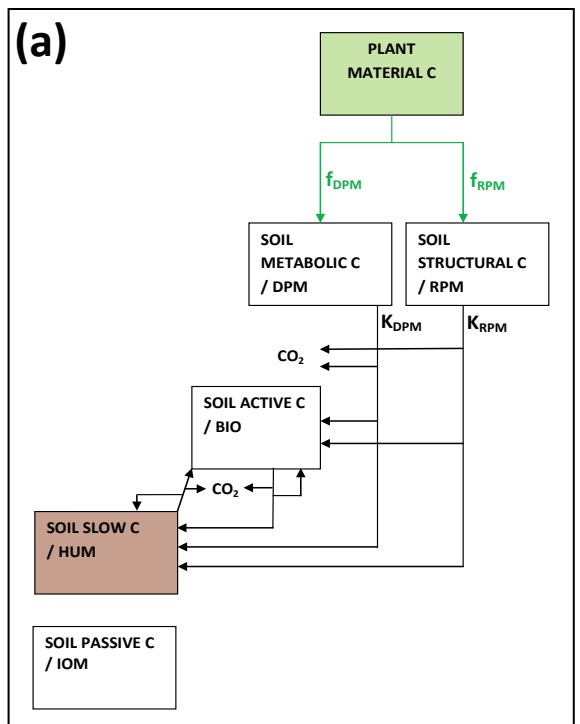
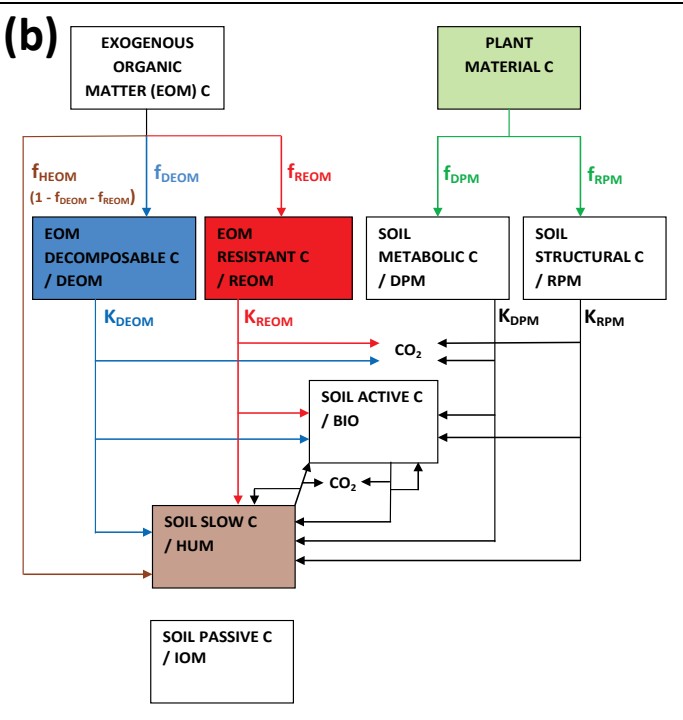





Figure 3

