# Peer review of "Modification of the RothC model to simulate soil C mineralization of exogenous organic matter"

_Biogeosciences, 2016_

## Referee Comment (RC1) · Anonymous Referee #1 · 3 Mar 2017

Modification of the RothC model to simulate soil C mineralization of exogenous organic matter

Claudio Mondini et al. MS No.: bg-2016-551

General comments

This could be a very good paper and the modification to RothC for organic amendments are very important, but I think adding three extra compartments to the model is not the best initial approach to take. The RothC model already allows for two types of material to enter the soil the first being "plant material" and the second "organic amendments". While the standard version of the model only allows for farmyard manure, there

is no reason why the authors, like Falloon (2001), should not have changed the proportions entering the DPM, RPM, and HUM pools to better model other types of organic amendment. By adding three extra compartments, will lead to a lower RMSE, due to increasing the number of pools, but this means increasing the number of parameters needed by the model, and making it far more complicated.

The authors should have initially used the approach used by Falloon, and for each different organic amendment they should have started by changing the default parameters for FYM from DPM 49%, RPM 49% and Hum 2%, to obtain more suitable proportions entering the DPM, RPM and Hum pools. This will give a lower RSME, and will also keep the model simple, but if by changing the proportion of organic amendment entering the DPM, RPM, and Hum pools, the model still does not give accurate predictions, the authors could then consider adding one or more extra pools.

So while the modification to RothC for organic amendments are very important, I do not think the authors have chosen the best approach.

---

## Author Comment (AC1) · 7 Mar 2017

We would like to thank the Referee #1 for her/his valuable and pertinent comments to the manuscript. Please find below our response to the comments.

Comments from Anonymous Referee #1:

This could be a very good paper and the modification to RothC for organic amendments are very important, but I think adding three extra compartments to the model is not the best initial approach to take. The RothC model already allows for two types of material to enter the soil the first being "plant material" and the second "organic amendments". While the standard version of the model only allows for farmyard manure, there

is no reason why the authors, like Falloon (2001), should not have changed the proportions entering the DPM, RPM, and HUM pools to better model other types of organic amendment. By adding three extra compartments, will lead to a lower RMSE, due to increasing the number of pools, but this means increasing the number of parameters needed by the model, and making it far more complicated. The authors should have initially used the approach used by Falloon, and for each different organic amendment they should have started by changing the default parameters for FYM from DPM 49%, RPM 49% and Hum 2%, to obtain more suitable proportions entering the DPM, RPM and Hum pools. This will give a lower RSME, and will also keep the model simple, but if by changing the proportion of organic amendment entering the DPM, RPM, and Hum pools, the model still does not give accurate predictions, the authors could then consider adding one or more extra pools. So while the modification to RothC for organic amendments are very important, I do not think the authors have chosen the best approach.

Answer:

We agree with the Referee's comment in that it perfectly describes the procedure we have followed in the modification of the model. As a matter of fact, in the initial stage of the study we tried to fit the respiratory curves of incubated soil by only modifying the partitioning factors of DPM, RPPM and HUM of added EOM and leaving unchanged the standard decomposition rates of the model (i.e. 10, 0.3 and 0.02 y-1 for DPM, RPM and HUM, respectively), as suggested by the Referee and carried out by Falloon (2001). This was performed trying to balance model simplicity and accuracy. Unfortunately, results of the optimization performed by changing only EOM pools partitioning factors were not satisfactory for most of the cases. As an example mean RMSE for household waste compost for different soils and incubation conditions was 20.0%, while for bioethanol residue it was 29.9%. Therefore, we further modified the model by introducing the possibility to vary the decomposition rate of EOM pools. As reported in the manuscript this model configuration allowed for a more accurate predictions of

respiratory curves (mean RMSE 4.5%). The above procedure followed to modify the model has already been reported in the current manuscript at the beginning of section "4.3 Model Optimization", LL 357 - 366. In the text we made reference to the work of Peltre et al. (2012), but the approach selected by these authors (i.e. changing only the partitioning factors of EOM pools) is the same as for Falloon (2001). We have now added reference to the latter in the manuscript. We have also added some sentences in the introduction section supporting the improvement in terms of simulation accuracy of an approach considering EOM specific decomposition rates.

Sentences added to the manuscript:

1 Introduction

L99

This model behaviour contrasts with the large variability in the decomposition rate of different EOMs and the evidence that model predictions can be improved by identification of EOM specific decomposition rate, as demonstrated by Mueller et al. (2003) with the DAISY model. Similarly to RothC, the original DAISY model involves two pools of added EOM with decomposition rates that are constant for a wide range of added organic materials. Muller et al. (2003) showed that adjusting the decomposition rates for each EOM significantly increased the model capacity to predict C mineralization in amended soils.

References:

Falloon, P.: Large scale spatial modelling of soil organic carbon dynamics, Ph.D. thesis, School of Life and Environmental Sciences, University of Nottingham, 2001.

Mueller, T., Magid, J., Jensen, L. S., and Nielsen, N. E.: Decomposition of plant residues of different quality in soil-DAISY model calibration and simulation based on experimental data, Ecol. Model., 166, 3-18, 2003.

Peltre, C., Christensen, B.T., Dragon, S., Icard, C., and Katterer, T.: RothC simulation

of carbon accumulation in soil after repeated application of widely different organic amendments, Soil Biol. Biochem., 52, 49-60, 2012.

---

## Referee Comment (RC2) · Anonymous Referee #2 · 5 Apr 2017

This manuscript presents a modification of the RothC model to simulate the addition and subsequent dynamics of exogenous organic matter. I found compelling the arguments that motivated this change in model structure, but I feel the authors need to do a better job in convincing why this change is necessary. I will explain this better below. Also, the manuscript is relatively long, and in many places it deviates from the main point, which is the modification of the model structure. The manuscript deviates to other topics related to SOM modeling and short- versus long-term incubations that are somewhat irrelevant to the main point of the manuscript.

My main issue with the current version of this manuscript is that the authors do not formally show an analysis that justify the modification of the current structure of RothC.

The authors do mention some preliminary work (lines 335-336, 358-360) in this direction, but concrete results are not presented in the manuscript. I think it is very important that the authors show how a simple modification of the DPM/RPM ratio is not enough to predict C mineralization from incubations with added EOM. Further, I think it would be important that the authors provide an analysis of the added benefit of a more complex model structure through calculation of common indexes such as Akaike Information Criteria, which would be helpful in deciding if a more complex model structure with more parameters increases predicted power or whether the simpler more parsimonious model structure is preferred.

Two minor points:

- System of equations on page 7 is missing an equation for the IOM pool. I know that this pool doesn't do anything, but for consistency with figure 2, you should add this pool.

- The problem you describe in section 4.4.2 is formally called equifinality or non-identifiability. It is well discussed for this type of models in Sierra et al. (2015, Soil Biol Biochem 90:197).

---

## Author Comment (AC2) · 20 Apr 2017

We would like to thank the Referee #2 for her/his valuable and relevant comments to the manuscript. Please find below our response to the main concern raised by the referee about the current version of the manuscript.

Comment of Reviewer #2:

My main issue with the current version of this manuscript is that the authors do not formally show an analysis that justify the modification of the current structure of RothC. The authors do mention some preliminary work (lines 335-336, 358-360) in this direction, but concrete results are not presented in the manuscript. I think it is very important

that the authors show how a simple modification of the DPM/RPM ratio is not enough to predict C mineralization from incubations with added EOM. Further, I think it would be important that the authors provide an analysis of the added benefit of a more complex model structure through calculation of common indexes such as Akaike Information Criteria, which would be helpful in deciding if a more complex model structure with more parameters increases predicted power or whether the simpler more parsimonious model structure is preferred.

Authors' reply:

We agree with the referee concerning the importance to demonstrate the necessity of the proposed model modification. In fact, and as reported in the manuscript (LL 358-367), the initial idea of our study was to fit the respiratory curves of laboratory incubated soil by only modifying the partitioning factors of DPM, RPPM and HUM of added EOM and leaving unchanged the standard decomposition rates of RothC (i.e. 10, 0.3 and 0.02 y-1 for DPM, RPM and HUM, respectively), as carried out by Falloon (2001) and Peltre et al. (2012). However, results of the fitting procedure showed that RMSE was not satisfactory in most of cases, being higher than 20%. Therefore, we modified the model by introducing the possibility to vary the decomposition rate of decomposable and resistant pools of EOM and this model structure showed a more accurate predictions of respiratory curves (mean RMSE of 4.5%). Following referee's suggestion to demonstrate in an analytical way the benefit of a more complex model structure, we have performed the calculation of the AIC index (Symonds and Moussalli, 2011) from the results of the respiratory curves fitting of 2 different model structures:

1) Current RothC model: the model only allows for variation of partitioning factors for EOM pools (DPM, RPM and HUM), while decomposition rate constant are fixed (i.e. 10, 0.3 and 0.02 y-1 for DPM, RPM and HUM, respectively).

2) Modified RothC (proposed in the MS): the model allows for modification of partitioning factors and DPM and RPM decomposition rates of EOM.

In total we calculated the AIC index for more than 80 respiratory curves and for the two model structures and results clearly showed that the increase in model complexity due to the introduction of new parameters was justified by the significant improvement in the model performance. According to the threshold values, based on AIC, indicated by Snipes and Taylor (2014) to determine the level of evidence for selecting between different models, the modified model was decisively better than the current one for all the respiratory curves selected for AIC determination.

The rationale beyond the proposed modification arises from the consideration that the current Roth C model structure is not adequate to describe the mineralization dynamics of EOMs characterized by a huge variability in terms of structure and degradability. We consider that the model performance would be increased by the possibility to set specific decomposition rates for decomposable and resistant EOM pools. In this perspective, models with a similar complex structure as in the proposed modified RothC (5 different pools of C input to the soil and specific decomposition rates for decomposable and resistant EOM) have been already proposed and successfully validated for amended soils (NC-SOIL: Noirot-Cosson et al., 2016 - CN-SIM: Petersen et al., 2005; Cavalli and Bechini, 2012).

References

Cavalli, D., and Bechini, L.: Sensitivity analysis and calibration of CN-SIM to simulate the mineralisation of liquid dairy manures, Soil Biol. Biochem, 43, 1207-1219, 2011.

Falloon, P.: Large scale spatial modelling of soil organic carbon dynamics, Ph.D. thesis, School of Life and Environmental Sciences, University of Nottingham, 2001.

Noirot-Cosson, P. E., Vadour, E., Gilliot, J. M., Gabrielle, B., and Houot, S.: Modelling the long term effect of urban waste compost applications on carbon and nitrogen dynamics in temperate croplands. Soil Biol. Biochem., 52, 138-153, 2016.

Peltre, C., Christensen, B. T., Dragon, S., Icard, C., and Katterer, T.: RothC simulation

of carbon accumulation in soil after repeated application of widely different organic amendments, Soil Biol. Biochem., 52, 49-60, 2012.

Petersen, B. M., Berntsen, J., Hansen, S., and Jensen, L. S.: CN-SIM - a model for the turnover of soil organic matter. I: Long-term carbon and radiocarbon development, Soil Biol. Biochem., 37, 359-374, 2005.

Snipes, M., and Taylor, D. C.: Model selection and Akaike Information Criteria: an example from wine ratings and prices. Wine Economics and Policy 3, 3-9, 2014.

Symonds, M. R. E., and Moussalli, A.: A brief guide to model selection, multimodel inference and model averaging in behavioural ecology using Akaike's information criterion. Behav. Ecol. Sociobiol., 65, 13-21, 2011.

---

## Author Response (AR1)

**Point by point reply to the Editor comments**

Dear Dr. Kirsten Thonicke,

thank you for your valuable and helpful comments to our response to the reports of reviewer 1 and reviewer 2.

We thank the editor and the two referees for their comments that were helpful to improve the quality of the manuscript.

Please find below a point by point reply to the Editor comments.

**Editor Comment 1**

Your response to reviewer #1:

In your revised manuscript, please make sure that you explicitly describe the methods on how the partitioning the organic amendment entering DPM, RPM and HUM was tested in the laboratory and in the model. Please explicitly describe the effect on RMSE and why you decided to add more pools instead. Please discuss the effect of entering more pools on lowering RMSE and on increasing the number of required parameters. Your suggested modifications in the introduction are a good start, but the point reviewer 1 addresses here, needs to be considered in the other sections of the manuscript as well.

**Authors' Reply 1**

In the revised version of the manuscript we have modified different sections (material and methods, results, discussion, tables and figures) to better explain the procedure we have followed to optimize the model, present the results of model comparison and discuss the reasons for introducing new EOM pools.

In the materials and methods section we have described the two-steps procedure we have adopted to select the model structure to be utilized in the simulation of the respiratory curves:

- fitting of respiratory curves by only varying EOM partitioning factors (standard model). In
   the standard model the decomposition rates of resistant and decomposable organic matter
   pools are fixed and the same for the corresponding pools of plant residues and EOM.
- fitting of respiratory curves by varying both partitioning factors and decomposition rates of EOM (modified model). The modified model introduces the possibility to set specific decomposition rate for decomposable and resistant EOM pools that are different from those of resistant and decomposable plant material.

We have then performed the calculation of the corrected Aikake information criterion (AICc, AIC corrected for small sample sizes) and associated statistics (Delta AICc -  $\Delta$ AICc, logarithm of Evidence Ratio - LER) to assess if the higher complexity of the modified model is justified in terms of increased model precision. We presented  $\Delta$ AICc and LER values instead of AIC because

such statistics are better suited than AIC to evaluate the relative strength of evidence a model with respect to another one (Mazerolle et al., 2016).

In the results section we have presented the results of the fitting procedure with the two models in terms of RMSE,  $\Delta$ AICc and LER values.

In the discussion section we explained why we have decided to utilize the modified model characterized by the extra pools. This choice was basically based on:

- results of RMSE,  $\Delta$ AICc and LER for the two models
- previous research showing that model performance in amended soils is improved by setting specific EOM decomposition rates. We have cited references showing that decomposable and resistant EOM pools presents different decomposition rates with respect to decomposable and resistant plant materials and that setting different decomposition rates for EOM enhances the capacity of the C models to describe mineralization dynamics in amended soils.

**Editor Comment 2**

Your response to reviewer #2:

- 1 Please add the AIC results that you have obtained from the two model structures in the results and discussion section.
- 2 Your response regarding the proposed model modification to discuss the RothC mineralization dynamics. The response to this reviewer comment should be included in the discussion section as well.

**Authors' Reply 2**

- 1. We have introduced the calculation of AICc and associated statistics ( $\Delta$ AICc LER) in the material and methods section and presented and commented the  $\Delta$ AICc and LER results of model comparison in the result and in the discussion sections.
- The authors response to the comment of reviewer 2 concerning the better description of EOM mineralization dynamics provided by the proposed model modification has been included in the discussion of the revised manuscript (LL 422-443).

**Editor Comment 3**

Please provide a modified manuscript that addresses all above-listed points.

**Authors' Reply 3**

In the revised version of the manuscript we have focused on all the point raised by the editor.

**Reference**

Mazerolle, M.J.: Improving data analysis in herpetology: using Akaike's Information Criterion (AIC) to assess the strength of biological hypotheses. Amphibia-Reptilia 27, 169-180, 2006.

**List of relevant changes in the manuscript**

**Introduction**

We have introduced a sentence supporting the improvement in terms of simulation accuracy of an approach considering EOM specific decomposition rates

**Materials and Methods**

**2.2.2 Modification of the RothC model**

We have explained the procedure we have adopted to compare the standard and modified model structure.

We have introduced the calculation of AIC and derived statistics utilized to assess the relative merit of two models in describing respiratory curves

We have added an equation for IOM pool to the system of equations reported at page 7 as suggested by reviewer 2.

**Results**

We have presented the results of model comparison in terms of RMSE,  $\Delta \text{AIC}$  and LER

**Discussion**

We have discussed the results of model comparison supporting the modified model. The reliability of the modified model was also supported by references showing the inadequacy of RothC in amended soil and the model improvement by setting specific EOM decomposition rates. We have introduced in the test the term equifinality to identify the problem discussed in section 4.4.2 as suggested by reviewer 2.

**Tables**

We have introduced a new table (Table 6) with RMSE,  $\Delta$ AICc and LER values obtained in the model comparison in the preliminary stage of the research.

**Figures**

We have slightly modified the figure 2 to better visualize the differences between the standard and modified model.

**Modification of the RothC model to simulate soil C mineralization of exogenous organic matter**

Claudio Mondini1, Maria Luz Cayuela2, Tania Sinicco1, Flavio Fornasier1, Antonia Galvez3, Miguel Angel Sánchez-Monedero2

[revised manuscript text omitted]

- fDPM = partitioning factor for DPM; fDEOM = partitioning factor for DEOM; fREOM = partitioning factor for REOM.
  KDPM = decomposition rate for DPM; KRPM = decomposition rate for RPM; KBIO = decomposition rate for BIO; KHUM = decomposition rate for HUM; KDEOM = decomposition rate for DEOM; KREOM = decomposition rate for REOM.
  P = plant (crop residue) input; E = EOM input; α = transfer coefficient to BIO pool; β = transfer coefficient to HUM pool.
  The C flow of the standard and modified model is reported in Fig. 2.
- 245 An Excel version of the modified model was then utilized to perform model fitting of the same 86 respiratory curves previously simulated with the standard model. The procedure utilized was the same with the exception that model fitting was conducted by simultaneously changing partitioning factors ( $f_{DEOM}$ ,  $f_{REOM}$ ,  $f_{HEOM}$ ) and decomposition rate constants ( $K_{DEOM}$ ,  $K_{REOM}$ ) of the different pools of EOM, considering the following constraints in order to avoid biologically unrealistic parameter estimates:

250  $\frac{f_{DEOM} + f_{REOM} + f_{HEOM} = 1}{f_{HEOM} < 0.3 \text{ for anaerobic digestates and agro-industrial wastes}}$   $\frac{K_{REOM} > 0.15 \text{ y}^{-1}}{1}$

 $\underline{\mathbf{R}_{\text{REOM}} > 0.15 \text{ y}}$

kDEOM < 230 y-1

[revised manuscript text omitted]